# The atypical chemokine receptor ACKR3/CXCR7 is a broad-spectrum scavenger for opioid peptides

Max Meyrath[1,9], Martyna Szpakowska [1,9], Julian Zeiner[2], Laurent Massotte[3], Myriam P. Merz[1], Tobias Benkel [2,4], Katharina Simon[2], Jochen Ohnmacht [5,6], Jonathan D. Turner [1], Rejko Krüger[5,7], Vincent Seutin[3], Markus Ollert[1,8], Evi Kostenis [2] & Andy Chevigné [1✉]

Endogenous opioid peptides and prescription opioid drugs modulate pain, anxiety and stress by activating opioid receptors, currently classified into four subtypes. Here we demonstrate that ACKR3/CXCR7, hitherto known as an atypical scavenger receptor for chemokines, is a broad-spectrum scavenger of opioid peptides. Phylogenetically, ACKR3 is intermediate between chemokine and opioid receptors and is present in various brain regions together with classical opioid receptors. Functionally, ACKR3 is a scavenger receptor for a wide variety of opioid peptides, especially enkephalins and dynorphins, reducing their availability for the classical opioid receptors. ACKR3 is not modulated by prescription opioids, but we show that an ACKR3-selective subnanomolar competitor peptide, LIH383, can restrain ACKR3's negative regulatory function on opioid peptides in rat brain and potentiate their activity towards classical receptors, which may open alternative therapeutic avenues for opioid-related disorders. Altogether, our results reveal that ACKR3 is an atypical opioid receptor with cross-family ligand selectivity.

[1] Department of Infection and Immunity, Luxembourg Institute of Health (LIH), rue Henri Koch 29, L-4354 Esch-sur-Alzette, Luxembourg. [2] Molecular, Cellular and Pharmacobiology Section, Institute of Pharmaceutical Biology, University of Bonn, Nussallee 6, 53115 Bonn, Germany. [3] Neurophysiology Unit, GIGA Neurosciences, University of Liège, avenue de l'hopital, B-4000 Liège, Belgium. [4] Research Training Group 1873, University of Bonn, Bonn, Germany. [5] Luxembourg Centre for Systems Biomedicine (LCSB), University of Luxembourg, avenue du Swing 6, L-4367 Belvaux, Luxembourg. [6] Department of Life Sciences and Medicine, University of Luxembourg, avenue du Swing 6, L-4367 Belvaux, Luxembourg. [7] Transversal Translational Medicine, Luxembourg Institute of Health (LIH), rue Thomas Edison 1A-B, L-1445 Strassen, Luxembourg. [8] Department of Dermatology and Allergy Center, Odense Research Center for Anaphylaxis, University of Southern Denmark, 5000 Odense, Denmark. [9] These authors contributed equally: Max Meyrath, Martyna Szpakowska. ✉email: andy.chevigne@lih.lu

pioid receptors are G protein-coupled receptors (GPCRs) that play a central role in reward processing, euphoria, analgesia, stress, anxiety, and depression. The family consists of three classical receptors: mu (μ or MOR), delta (δ or DOR), and kappa (κ or KOR) and a fourth, non-classical nociceptin receptor (NOP, also known as orphanin FQ receptor)[1,2]. The classical receptors are activated by three major families of opioid peptides, namely endorphins, enkephalins, and dynorphins, each showing a preference for one or two families, while the non-classical NOP receptor shows a high affinity and selectivity towards nociceptin[3]. All endogenous opioid peptides derive from proteolytic cleavage of large protein precursors and are mainly produced in the central nervous system (CNS), but also in the adrenal and pituitary gland and by several types of immune cells[4,5]. With some exceptions, these ligands trigger downstream receptor signaling via G proteins, which is followed by arrestin recruitment, leading to receptor desensitization, and internalization[6,7]. Opioid receptors are established drug targets for non-peptide opioids such as morphine, fentanyl, or naloxone. These opioid receptor modulators are the most widely used analgesics in the clinic but their use is associated with severe drawbacks like tolerance, dependence, or respiratory depression, which were proposed to be linked to ligand and receptor bias towards arrestin recruitment[8–10]. Thus, a better understanding of opioid receptor signaling regulation and bias as well as new strategies to modulate opioid receptors with less adverse effects are not only timely but also urgently needed, especially considering the current opioid crisis.

Opioid receptor expression, signaling, and desensitization are furthermore influenced by their interactions with other GPCRs, notably chemokine receptors[11–13]. Chemokine receptors bind to chemokines, which are small (8–14 kDa) secreted chemoattractant cytokines, regulating cellular processes like migration, adhesion, and growth and thereby playing a crucial role in inflammatory and developmental processes[14,15]. To date, nearly 50 chemokines and 20 classical chemokine receptors have been identified in humans[16,17]. Similar to the opioid receptor–ligand network, many chemokine receptors recognize multiple chemokines, and, vice versa, many chemokines activate more than one receptor. Notably, within this network, a small subfamily of receptors, called atypical chemokine receptors (ACKRs), plays essential regulatory roles. ACKRs bind chemokines without triggering G protein signaling but instead participate in chemotactic events by transporting or capturing the chemokines or internalizing and degrading the ligands in order to resolve inflammatory processes or to shape chemokine gradients[18–20].

One such atypical chemokine receptor, ACKR3 (formerly CXCR7), is expressed in numerous regions of the CNS and in the adrenal glands but also on endothelial cells and diverse immune cells[21–23]. It plays crucial roles in neuronal and cardiovascular development and in the migration of hematopoietic stem cells[24,25]. The functions of ACKR3 are proposed to mainly rely on arrestin recruitment, which is indicative of ACKR3 activation, while its signaling capacity remains highly debated and may be cell-context-dependent[26–29]. ACKR3 binds two endogenous chemokines, CXCL12 and CXCL11, which also activate the classical chemokine receptors CXCR4 and CXCR3[30], respectively, as well as the virus-encoded CC chemokine vMIP-II/vCCL2[31]. Moreover, ACKR3 was described as the receptor for the non-chemokine ligands adrenomedullin[32] and MIF[33]. ACKR3 was also shown to bind to BAM22, a small proenkephalin-derived peptide in the adrenal glands, inducing direct anxiolytic-like behavior in mice[34]. Recently, ACKR3 was proposed to have a distinctive ligand-binding mode and activation mechanism, showing a particularly high propensity for arrestin recruitment and ligand internalization[35,36].

In this study, we show that ACKR3 is abundantly expressed in the same brain regions as the classical opioid receptors and that, besides BAM22, ACKR3 is activated by a large array of endogenous opioid peptides found in the CNS and immune cells, including those from the enkephalin, dynorphin and nociceptin families. However, contrary to the other four opioid receptors but in keeping with its atypical receptor features, ACKR3 is unable to activate canonical G protein signaling. Instead, it exclusively recruits arrestins in response to opioid peptides. We show that ACKR3 acts as a scavenger towards this family of neuromodulators, thus regulating their availability for signaling through the established opioid receptors, similarly to its role in chemokine gradient modulation. Hence, we propose ACKR3 as a promiscuous atypical opioid receptor (AOR) that functions as a scavenger receptor to regulate not only the abundance of chemokines but also of opioid peptides.

## Results

**ACKR3 is activated by a broad range of opioid peptides.** In a recent study, we suggested that the proenkephalin-derived peptide BAM22 shares structural and functional features important for ACKR3 binding and activation with the N terminus of chemokine ligands[35]. Given that all endogenous opioid peptides show remarkable sequence homologies including the F/YGGFL/M motif at their N termini, as well as several positively charged residues throughout the sequence (Table 1), we wondered whether BAM22 and the related peptides are the only opioid peptides able to activate ACKR3. Therefore, we screened a library of 58 opioid peptides (5 μM, Supplementary Table 1) for their ability to induce β-arrestin-2 recruitment to ACKR3 (indicative of ACKR3 activation), and, additionally to CXCR4 and CXCR3, two classical chemokine receptors sharing ligands with ACKR3, which served as negative controls. Besides BAM22, BAM18, and Peptide E previously reported as ACKR3 ligands[34], our screening revealed that numerous other opioid peptides are capable of inducing β-arrestin-2 recruitment to ACKR3. These included adrenorphin, another proenkephalin-derived peptide, but also peptides from the nociceptin and dynorphin families (Fig. 1a). Endorphins and endomorphins, however, did not activate ACKR3. None of these peptides acted as ACKR3 antagonist (Supplementary Fig. 1a) or induced β-arrestin-2 recruitment to CXCR4 or CXCR3 (Fig. 1a).

We then sought to further characterize the interactions of the opioid peptides with ACKR3 to establish whether their activity towards this receptor may be of physiological relevance. To this end, we performed pharmacological analysis, investigating the potency and efficacy of the different hits towards ACKR3 as well as the classical opioid receptors in β-arrestin-1 and β-arrestin-2 recruitment (Fig. 1b–f, Table 1, Supplementary Fig. 1b). ACKR3 was activated by several endogenous opioid peptides such as dynorphin A, dynorphin A 1–13, big dynorphin, BAM22 or adrenorphin at low concentrations comparable to their activity on the classical opioid receptors. Higher concentrations of dynorphin B, nociceptin or nociceptin 1–13-amide were necessary for ACKR3 activation. Surprisingly, ACKR3 was also fully activated by the weak partial NOP agonist Phe1ψ(CH$_2$-NH)-Gly2-nociceptin-1–13 amide (FψG nociceptin 1–13) as well as by endogenous truncated dynorphin variants, dynorphin 2–13 and dynorphin 2–17 (Table 1), which do not activate the classical opioid receptors but were shown to have a physiological effect[37,38]. However, ACKR3 seems to show a certain degree of selectivity as several peptides, like endorphins, short endomorphins and leu- or met-enkephalin did not trigger β-arrestin recruitment. This was further confirmed in different cellular backgrounds, such as HEK293T and CHO-K1 (Supplementary Fig. 1c, d) and in binding competition studies, showing that all of

**Table 1 Sequences of opioid peptides and their activity in β-arrestin-1 recruitment.**

| Name | Sequence | ACKR3 $EC_{50}$ nM - Max % | MOR $EC_{50}$ nM - Max % | DOR $EC_{50}$ nM - Max % | KOR $EC_{50}$ nM - Max % | NOP $EC_{50}$ nM - Max % |
|---|---|---|---|---|---|---|
| Dynorphin A | **YGGF**LRRIRPKLKWDNQ | 110 (94.7-129)-119 | ~1500-66 | 240 (115-2143)-76 | 12.7 (10.5-15.2)-100 | NA-6 |
| Dynorphin A 2-17 | -**GGF**LRRIRPKLKWDNQ | ~2000-87[a] | NA-0 | NA-0 | NA-0 | NA-0 |
| Dynorphin A 2-13 | -**GGF**LRRIRPKLK-NH2 | ~6000-83[a] | NA-0 | NA-0 | NA-0 | NA-0 |
| Dynorphin A 1-13 | **YGGF**LRRIRPKLK | 61.9 (50.4-76.1)-94 | ~2000-67 | 319 (188-1038)-81 | 2.5 (1.5-3.7)-93 | NA-6 |
| Dynorphin B | **YGGF**LRRQFKVVT | 727 (385-8608)-103 | ~2000-53 | 158 (101-331)-90 | 10.9 (6.3-17.2)-99 | NA-3 |
| Leumorphin | **YGGF**LRRQFKVVTRSQEDPNAYSGELFDA | 1320 (841-3050)-96[a] | ND-17 | ~3000-53 | 21.4 (17.4-26.4)-88 | NA-8 |
| Big Dynorphin | **YGGF**LRRIRPKLKWDNQKR**YGGF**LRRQFKVVT | 108 (89.3-133.9)-127 | 652 (502-966)-92 | 529 (251-8197)-97 | 19.1 (12.9-30.6)-138 | ND-11 |
| Adrenorphin | **YGGF**MRRV-NH2 | 56.5 (36.6-95.0)-96 | 41.6 (30.7-57.3)-96 | 157 (104-310)-96 | 43.5 (33.7-57.9)-98 | NA-3 |
| BAM22 | **YGGF**MRRVGRPEWWMDYQKRYG | 23.5 (11.0-53.1)-112 | 10.0 (7.1-13.8)-100 | 367 (193-2930)-95 | 44.2 (33.6-58.2)-111 | NA-2 |
| Met enkephalin | **YGGF**M | NA-0[a] | 419 (350-528)-95 | 16.3 (12.0-21.6)-100 | NA-2 | NA-0 |
| Nociceptin | F**GGF**TGARKSARKLANQ | >10,000-49[a] | NA-0 | NA-0 | NA-1 | 19.6 (13.4-28.5)-100 |
| Nociceptin 1-13 | F**GGF**TGARKSARK-NH2 | ~10,000-63[a] | NA-0 | NA-0 | NA-2 | 15.7 (10.7-22.1)-107 |
| FΨG nociceptin 1-13 | [FΨ(CH2-NH)G]GFTGARKSARK-NH2 | 2966 (1660-22,000)-95[a] | NA-0 | NA-0 | NA-2 | 56.5 (34.9-106.7)-18 |
| Endomorphin-1 | YPWF-NH2 | ND-20[a] | 216 (111-8640)-95[a] | NA-0 | NA-0 | NA-0 |
| Endomorphin-2 | YPFF-NH2 | ND-11[a] | 231 (124-876)-80[a] | NA-0 | NA-0 | NA-0 |
| β-endorphin | **YGGF**MTSEKSQTPLVTLFKNAIIKNAYKKGE | ND-13[a] | ~1000-85[a] | 441 (339-679)-102 | ND-14 | NA-0 |
| LIH383 | F**GGF**MRRK-NH2 | 0.61 (0.19-1.17)-99 | NA-1 | NA-1 | NA-9 | NA-1 |

$EC_{50}$ values are indicated in nanomolar (nM) with 95% confidence interval (CI).
Max: maximum signal measured at 3 μM expressed as % of the positive control/reference peptide.
ND: Not determinable since saturation was not reached.
NA: No activity or activity below 10% of positive control in the concentration range tested.
YGGF motif conserved in most of the opioid peptides is bold.
Full name of FΨG nociceptin 1–13: [Phe1Ψ(CH2-NH)-gly2]nociceptin-(1-13)-NH2.
[a]Measured at 9 μM.

the identified ligands were able to compete with, and displace Alexa Fluor 647-labeled CXCL12 from ACKR3 (Fig. 1g, Supplementary Table 2).

These data reveal that ACKR3 is selectively activated by various endogenous opioid peptides from different families (Fig. 1h) in a concentration range similar to that observed for activation and signaling via the long-established opioid receptors, strongly pointing to a physiological relevance of these newly identified ligand–receptor interactions.

**ACKR3 is the only chemokine receptor activated by opioid peptides**. Just like classical opioid receptors, many chemokine receptors have multiple ligands, which they often share with other receptors[16]. Thus, we wondered whether ACKR3 is the only member of the chemokine receptor family activated by opioid peptides. To this end, we tested all chemokine receptors for arrestin recruitment in response to the different ACKR3-recognizing opioid peptide ligands at a saturating concentration using the same Nanoluciferase complementation assay. None of the peptides induced similar β-arrestin-1 or β-arrestin-2 recruitment to any of the 21 other chemokine receptors (Fig. 2 and Supplementary Fig. 2a). A weak induction of β-arrestin recruitment was detectable towards several receptors such as CCR3, CXCR3, and CX3CR1 treated with big dynorphin. However contrary to ACKR3 or opioid receptors, their responses to big dynorphin were severely reduced compared to those achieved with their cognate chemokines (Supplementary Fig. 2b and Supplementary Table 3). These data provide strong support for the notion that the capacity to recruit arrestins in response to endogenous opioid peptides is unique and distinguishing for ACKR3 among all chemokine receptor family members.

**Ligand SAR study reveals mixed ACKR3 opioid-binding pocket**. The activity and selectivity of opioid peptides are generally proposed to be supported by distinct regions of the peptide. While the amino-terminal YGGF core, the message, is responsible for receptor activation through interactions with the binding pocket, the address, composed of the carboxy-terminal residues beyond the YGGF core, provides the structural basis for receptor selectivity. While the YGGF-L/M sequence is necessary and sufficient for high-affinity binding and modulation of DOR and MOR, this does not hold true for KOR[39] or ACKR3 as observed

in our initial screening, where neither met- nor leu-enkephalin (YGGF-L/M) activated ACKR3.

To gain further insights into the binding and activation modes of ACKR3 compared to classical opioid receptors, we performed a structure-activity relationship (SAR) study based on the octapeptide adrenorphin (YGGFMRRV-NH2, formerly metorphamide) (Fig. 3a). Adrenorphin triggered arrestin recruitment to ACKR3, MOR, DOR, and KOR with roughly the same potency (Fig. 3b), providing a suitable base for investigating the activation mode of the four receptors. We performed an alanine scan of adrenorphin and introduced substitutions by closely related amino acids or other modifications such as N- and C-terminal extension, D-amino acid replacement or dimerization. We evaluated the ability of these modified peptides to activate ACKR3 and the opioid receptors in β-arrestin-1 recruitment assay (Fig. 3a). Interestingly, the message and address sequences were somewhat different for ACKR3 compared to classical opioid receptors, despite a similar trend in potency changes. ACKR3 appears to be more tolerant to modifications of the N-terminal tyrosine residue, critical for the activation of classical opioid receptors. Indeed, variants displaying a leucine or a phenylalanine retained the parental activity, with Y1F mutation, mimicking the nociceptin peptide N terminus, resulting in a tenfold improvement in potency (Figs. 3a, c). However, similarly to classical receptors, the phenylalanine at position 4 of the YGGF-L/M core was found to be crucial for ACKR3 binding, as any mutation, with the exception of F4W, was detrimental for receptor activation. Methionine-to-leucine substitution at position 5, mimicking peptides of the dynorphin family, improved binding to KOR but significantly reduced the binding to MOR and ACKR3 (Fig. 3d), whereas mutations at positions 6 in the di-arginine motif abolished the activity towards KOR and ACKR3, but largely improved the activity towards MOR (Fig. 3e).

Based on the above SAR analysis we concluded that the interaction mode of ACKR3 with opioid peptides is in some aspects distinct from that of the classical opioid receptors, while at the same time ACKR3 shares important interaction determinants with all of the other four opioid receptors. This feature likely provides a molecular explanation why ACKR3 binds and responds to opioid peptides from different families.

**ACKR3 is unresponsive to alkaloid opioids and opioid drugs**. Based on its ability to respond to different families of endogenous

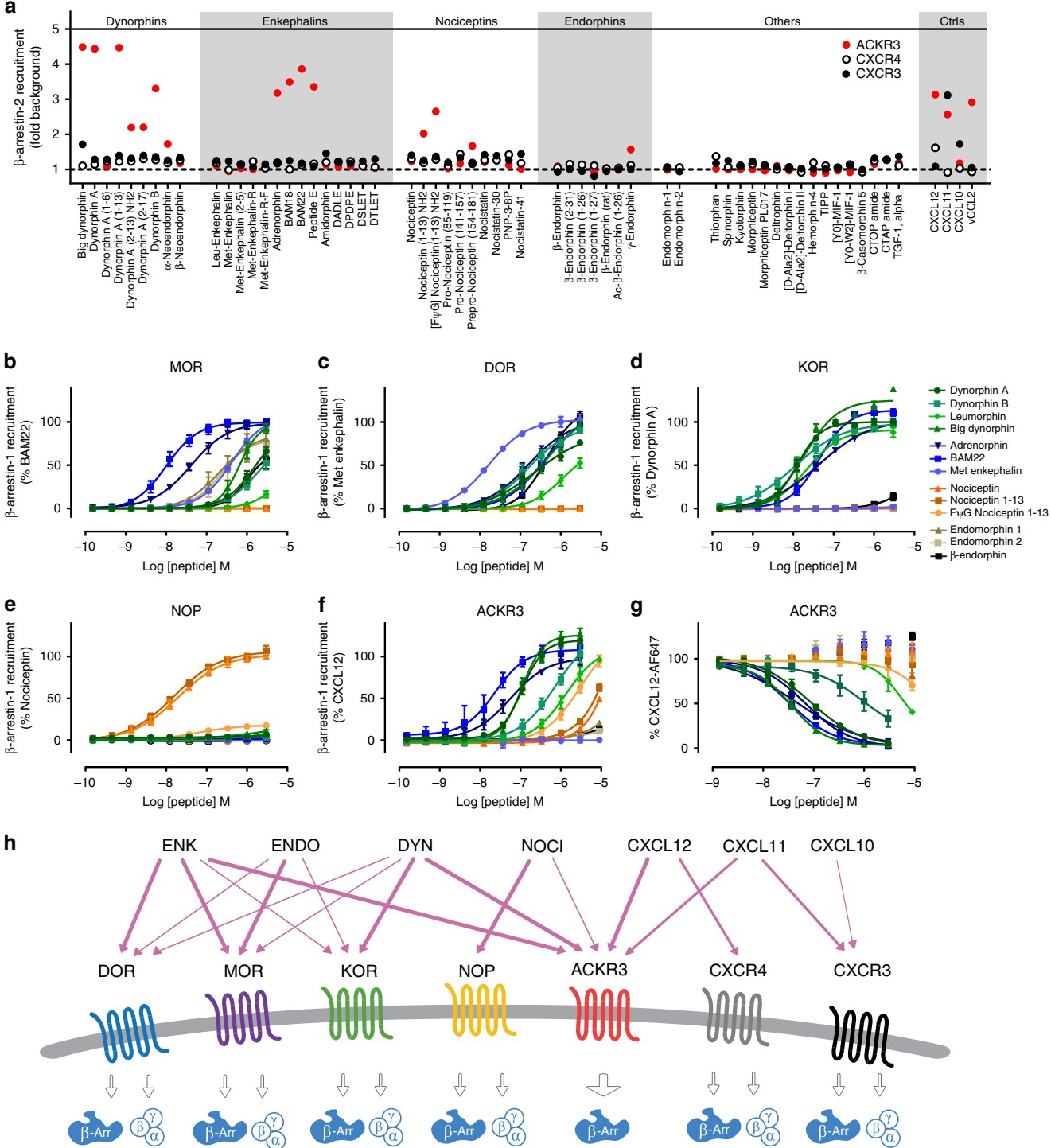

**Fig. 1 Opioid-peptide library screening on ACKR3 and hit activity comparison. a** The ability of 58 compounds, including natural opioid peptides from the four opioid families, variants thereof and small molecule opioid receptor modulators, to induce β-arrestin-2 recruitment to ACKR3, CXCR4 and CXCR3 in U87 cells at a concentration of 5 μM. For full peptide names and sequences, see Supplementary Table 1. Chemokine positive controls (Ctrls) were used at a concentration of 300 nM. Results are expressed as fold change over vehicle-treated cells and presented as mean of two technical replicates for ACKR3 and CXCR4. A single measurement was performed for CXCR3. **b–f** Comparison of potency and efficacy of ACKR3-activating and other representative opioid peptides in inducing β-arrestin-1 recruitment to the opioid receptors MOR (**b**), DOR (**c**), KOR (**d**), NOP (**e**), and ACKR3 (**f**) in U87 cells. Results are expressed as percentage of indicated agonist response. The corresponding EC$_{50}$ and Emax values are summarized in Table 1. **g** Binding competition of ACKR3-activating and other representative opioid peptides with Alexa Fluor 647-labeled CXCL12 (5 nM) on U87-ACKR3 cells determined by flow cytometry. Results from b-g are presented as mean ± S.E.M of three or four independent experiments (*n* = 3 or 4). Peptides from the enkephalin, dynorphin, nociceptin and endorphin families are depicted in blue, green, orange and gray scale, respectively. **h** Schematic representation of the interactions of the four opioid receptors and chemokine receptors with their respective ligands. The newly identified opioid peptide pairings with ACKR3 are shown, highlighting the cross-family selectivity of ACKR3. Source data are provided as Source Data file.

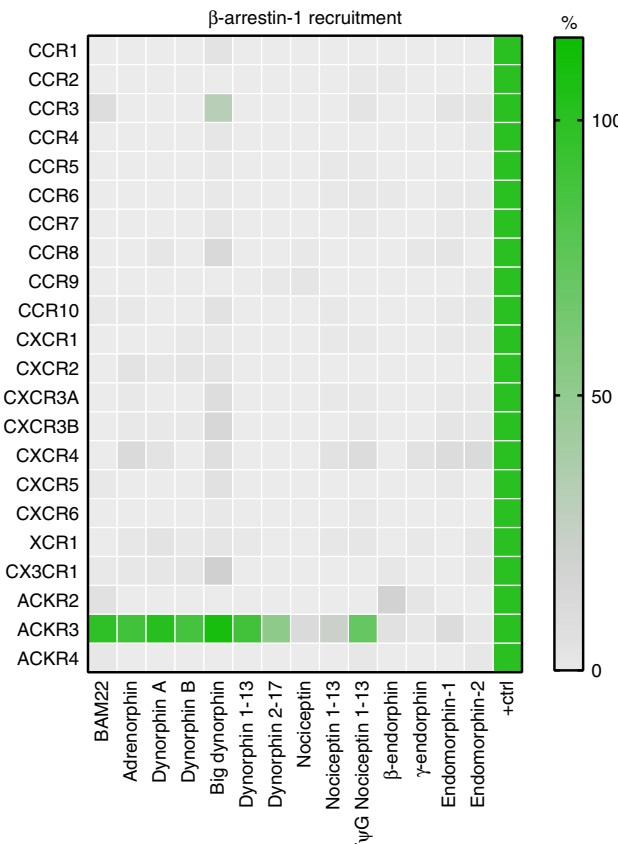

**Fig. 2 Specific activation of ACKR3 by opioid peptides.** Agonist activity of opioid peptides (3 μM) representative of the four opioid families towards 18 classical and three atypical chemokine receptors evaluated in β-arrestin-1 recruitment assay in U87 cells. Results are expressed as percentage of signal monitored with saturating concentration of active chemokines (100 nM) used as positive controls (Supplementary Table 3) and presented as mean of three independent experiments ($n = 3$). Source data are provided as a Source Data file.

opioid peptides and the binding mode similarities with classical opioid receptors, we wondered whether ACKR3 could also respond to non-endogenous opioid ligands commonly used to activate or inhibit the classical opioid receptors. Besides prototypical opioid tool compounds like D-Ala$^2$,D-Leu$^5$-Enkephalin (DADLE) or [D-Ala$^2$, $N$-MePhe$^4$, Gly-ol]-enkephalin (DAMGO), we tested approved pain medications such as morphine, fentanyl or buprenorphine in β-arrestin recruitment assays. All molecules showed their expected agonist or antagonist activities on their respective opioid receptors (Fig. 4a). For morphine, only a weak β-arrestin recruitment to MOR was observed in line with previous reports[40]. At high concentrations, many of these molecules, although designed to target specifically one receptor, showed some activity towards other opioid receptors, similar to the endogenous ligands. However, ACKR3 was not responsive to any of the molecules tested, even at high concentrations. Weak activation of ACKR3 was observed with the KOR agonist U50488 and antagonist nor-binaltrophimine, but with a much weaker potency compared to KOR.

These results show that although ACKR3 shares several endogenous opioid peptide ligands with classical opioid receptors, it is not modulated by opiate analgesics or synthetic opioid drugs targeting classical opioid receptors. This is in accordance with the absence of the binding pocket determinants in ACKR3 that are

required for productive interaction with such ligands (Supplementary Fig. 3).

**LIH383 is a highly selective subnanomolar agonist of ACKR3.** Despite its attractiveness as drug target, there is still a clear lack of small, pharmacologically well-characterized and easily available molecules for specific ACKR3 modulation in vitro or in vivo. To overcome this limitation, we took advantage of the adrenorphin SAR data (Fig. 3a) with the aim of developing a highly potent and selective ACKR3 modulator. We designed a second generation of peptides, using the adrenorphin Y1F variant as scaffold as it showed a 10-fold increase in potency towards ACKR3 and over 100-fold reduction of potency towards the classical opioid receptors MOR, DOR, and KOR as compared to WT adrenorphin (Figs. 3a, c). Other mutations increasing the potency towards ACKR3 including F4W, V8F, V8K, or 9R were further combined and the resulting peptides were tested in a β-arrestin recruitment assay (Supplementary Tables 4, 5). Of all the combinations tested, the octapeptide FGGFMRRK-NH$_2$ (designated LIH383) was the most potent ACKR3 agonist. It competed directly with CXCL12-AF647 for ACKR3 binding at low nanomolar concentrations (Supplementary Fig. 4a) and was more potent in inducing β-arrestin recruitment to ACKR3 (EC$_{50}$ = 0.61 nM) than the full-length chemokine ligands CXCL12 or CXCL11 (EC$_{50}$ = 1.2 nM and 2.2 nM, respectively) (Fig. 4b). Importantly, no activation or inhibition of any other opioid receptor, nor of any other chemokine receptor could be detected upon LIH383 treatment, even at concentrations as high as 3 μM (Figs. 4c, d, Supplementary Fig. 4b, c). Remarkably, LIH383 had equivalent activity on human and mouse ACKR3 (mACKR3) (Figs. 4b, e). Moreover, fluorescent labeling at the C-terminal lysine of LIH383 did not alter its binding properties as shown by Cy-5-labeled LIH383 binding to ACKR3-expressing U87 cells, but not to native or CXCR4-expressing U87 cells (Fig. 4f). Therefore, LIH383 is a highly attractive and versatile tool for specific ACKR3 modulation or detection of ACKR3-expressing cells in human and rodent models and is particularly suitable for investigating the biological consequences of opioid peptide interactions with ACKR3[29,41].

**ACKR3 does not signal in response to opioid peptides.** To decipher the function and impact of ACKR3 on the opioid system, we tested the ability of opioid peptides to trigger downstream signaling through ACKR3 in U87 cells that have no endogenous expression of CXCR4[31]. We first applied a whole-cell optical biosensing approach based on dynamic mass redistribution (DMR), which enables to detect multiple downstream signaling events including all four major G protein pathways[42–44]. In agreement with other studies[45–50], we did not detect any ACKR3-dependent signaling upon chemokine stimulation (Fig. 5a). Likewise, no difference in DMR signal was observed between ACKR3-transfected and non-transfected cells in response to opioid peptides (Fig. 5b–d and Supplementary Fig. 5a, b). In contrast, U87 cells expressing the G protein signaling-competent CXCR4, KOR, or NOP, did show a strong DMR in response to CXCL12, dynorphin A/adrenorphin or nociceptin 1–13, respectively, in accordance with a robust activation of downstream signaling pathways by these receptors (Fig. 5a–d). In line with these observations, we did not detect any interaction of ACKR3 with mini G (mG) proteins (mGi, mGs, mGq, or mG12/13) upon chemokine or opioid peptide treatment, in contrast to classical opioid receptors, which all efficiently recruited mini Gi (Fig. 5e and Supplementary Fig. 5d). Moreover, ERK phosphorylation levels monitored by homogeneous time-resolved fluorescence (HTRF) remained unchanged upon ligand stimulation of cells stably expressing ACKR3, whereas a strong

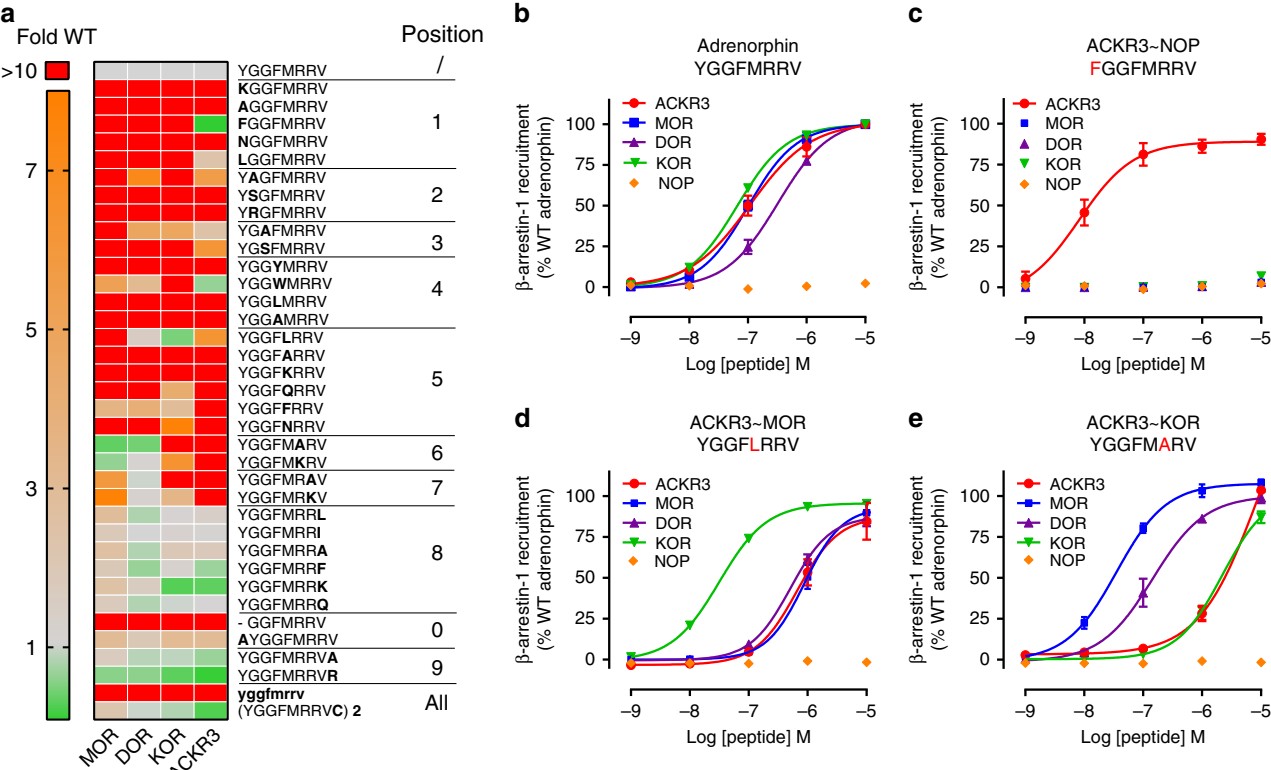

**Fig. 3 SAR analysis of adrenorphin variants on ACKR3 and classical opioid receptors. a** Comparison of the impact of substitutions, truncations, extensions, D-amino acid replacement or dimerization on the agonist activity of adrenorphin towards ACKR3, MOR, DOR, and KOR. The agonist activity of each variant was evaluated in β-arrestin-1 recruitment assay in U87 cells and expressed as fold change over the activity of wild-type adrenorphin. None of the peptides showed agonist activity towards NOP. **b–e** Comparison of potency and efficacy of adrenorphin (**b**) and its variants bearing mutations Y1F (**c**), M5L (**d**) and R6A (**e**) to induce β-arrestin-1 recruitment to ACKR3 and the opioid receptors KOR, MOR, DOR and NOP in U87 cells. ~ indicates a similar impact of the modification on the potency of the peptide towards ACKR3 and the indicated opioid receptor. Results represent the mean (**a**) or mean ± S.E.M (**b–e**) of three independent experiments ($n = 3$). Corresponding $EC_{50}$ values are available in Supplementary Table 4. Source data are provided as a Source Data file.

increase in ERK phosphorylation was observed between 2 and 120 min after CXCL12 stimulation of cells stably expressing CXCR4 (Fig. 5f) with total ERK levels remaining unchanged (Supplementary Fig. 5c). These results were further corroborated by the absence of activation of the MAPK/ERK-dependent Serum Response Element (SRE) upon opioid peptide or chemokine stimulation in ACKR3-positive U87 cells (Fig. 5g, left panel) or HEK293T and CHO-K1 cells (Supplementary Fig. 5e), in contrast to the robust signal increase in CXCR4- or classical opioid receptor-expressing cells upon stimulation with the respective ligands. Similar absence of signaling through ACKR3 in response to opioid or chemokine ligands was shown for calcium-dependent Nuclear Factor of Activated T-cell Response Element (NFAT-RE) activation (Fig. 5g, right panel).

These data demonstrate that opioid peptides induce β-arrestin recruitment to ACKR3 without triggering signaling typical to GPCRs suggesting that ACKR3 may act as a scavenger for opioid peptides in a manner akin to that observed for its chemokine ligands.

**ACKR3 mediates efficient uptake of various opioid peptides.** To investigate the ability of ACKR3 to scavenge opioid peptides, we first measured the uptake of fluorescently labeled opioid peptides of different families by cells expressing ACKR3 or the corresponding classical opioid receptors using imaging flow cytometry.

For dynorphin A (1–13), a clear intracellular accumulation of the fluorescently labeled peptide after 40-min stimulation could

be observed in U87-ACKR3 cells, with a notably higher number of distinguishable vesicle-like structures and mean fluorescent intensity compared to U87 cells or U87-ACKR3 cells pre-incubated with LIH383 at a saturating concentration (Fig. 6a), demonstrating that ACKR3 can mediate the uptake of opioid peptides. Moreover, the uptake of dynorphin A (1–13) by ACKR3 was more efficient compared to that of KOR, the main classical opioid receptor for this peptide, despite the lower potency of dynorphin A (1–13) towards ACKR3 (Fig. 6b and Table 1). Similar observations were made for labeled big dynorphin, the precursor of dynorphin A and B, and for BAM22, a peptide from the enkephalin family. Indeed, despite its similar potency towards the two receptors, BAM22 was markedly more internalized by ACKR3-positive than by MOR-positive cells. The low-affinity ligand nociceptin was also internalized by ACKR3 to a degree equivalent to the corresponding classical opioid receptor NOP (Fig. 6b). Importantly, this ACKR3-driven intracellular accumulation of opioid peptides was also associated with a reduction of their availability in the extracellular space. For instance, we found that the apparent potency of dynorphin A in inducing KOR activation was reduced in the presence of ACKR3-expressing cells. This effect was reversed when ACKR3-expressing cells were pretreated with LIH383 or CXCL12 but not LIH383ctrl or irrelevant chemokine CXCL10, illustrating the plausible scavenging function of ACKR3 (Fig. 6c and Supplementary Fig. 6).

In line with this scavenging function, ACKR3 showed an atypical cellular localization, internalization and trafficking

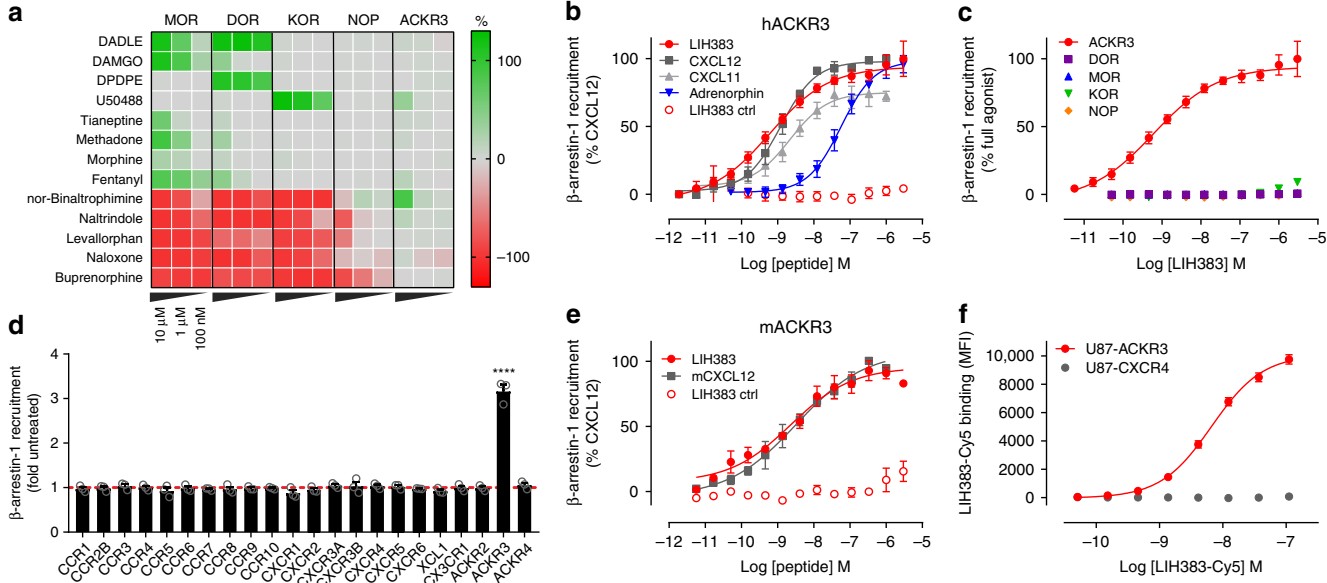

**Fig. 4 Activity of opioid modulators and development of LIH383 as ACKR3-selective agonist. a** Agonist (green color scale) and antagonist (red color scale) activity towards ACKR3 of opioid modulators (10 μM, 1 μM, and 100 nM) commonly used for research purposes or in clinic and comparison with the opioid receptors MOR, DOR, KOR, and NOP monitored in a β-arrestin-1 recruitment assay. Antagonist activity was measured following addition of BAM22 (50 nM), met-enkephalin (70 nM), dynorphin A (50 nM), nociceptin (70 nM) and CXCL12 (4 nM) on MOR, DOR, KOR, NOP, and ACKR3, respectively (**b**, **e**) Agonist activity of LIH383 (FGGFMRRK) towards human (**b**) and mouse (**e**) ACKR3 and comparison with other endogenous ACKR3 chemokine ligands, and a control peptide (MRRKFGGF), consisting of the eight amino acids building LIH383 in a different arrangement. **c**, **d** LIH383 selectivity evaluated by comparison of β-arrestin-1 recruitment to ACKR3 and the opioid receptors MOR, DOR, KOR and NOP (**c**) or all the other chemokine receptors known to recruit β-arrestin (3 μM) (**d**). **f** Selective binding of fluorescently labeled LIH383 (LIH383-Cy5) to ACKR3-expressing cells. All assays were performed in U87 cells. Results represent the mean (**a**) or mean ± S.E.M (**b**–**f**) of three to five independent experiments ($n = 3$–5) except for LIH383 on hACKR3 ($n = 9$). All fitted values are available in Supplementary Table 6. For statistical analysis, one-way ANOVA with Bonferroni multiple comparison test was used. ****$p < 0.0001$. Source data and statistical analysis parameters are provided as Source Data file.

pattern compared to classical opioid receptors. In agreement with previous reports[35,50], we observed that a much higher proportion of ACKR3 was present intracellularly compared to the cell surface. In contrast, classical opioid receptors MOR, DOR, KOR, and NOP were mainly localized at the plasma membrane (Fig. 6d). Moreover, despite an efficient uptake of various opioid peptides and their delivery to the early endosomes (Fig. 6e), the overall reduction of ACKR3 at the cell surface was much less pronounced than for classical opioid receptors, likely reflecting the rapid cycling of ACKR3 between the plasma membrane and intracellular compartments (Figs. 6f, d). Similar to what was reported for chemokines[50,51], we found that agonist removal after stimulation led to progressive increase of ACKR3 at the plasma membrane, while for classical receptors like KOR such recovery was not observed (Fig. 6g). This was further corroborated by results obtained with bafilomycin A1, an inhibitor of vacuolar-type H$^+$-ATPases. Previous studies showed that low endosomal pH is needed for chemokine dissociation from ACKR3 and efficient receptor recycling and resensitization[50,51]. We observed that treatment with bafilomycin A1 resulted in decreased receptor recovery at the plasma membrane following stimulation with diverse opioid peptides, whereas it had no effect on surface levels of KOR (Fig. 6h).

Altogether, these results demonstrate that ACKR3 can support a rapid and efficient uptake of opioid peptides of different families through continuous receptor cycling between the intracellular compartments and the plasma membrane, leading to a progressive depletion of extracellular opioid peptides, thereby limiting their availability for classical receptors.

**ACKR3 regulates the availability of opioid peptides in CNS.** To support a physiological relevance of the observed opioid peptide scavenging capacity of ACKR3, we then analyzed, using the Brainspan database (www.brainspan.org), its gene expression profile in comparison with classical opioid receptors in different brain regions corresponding to important centers for opioid activity. In agreement with previous studies[22,52], we found that not only was *ACKR3* expressed in many of these regions such as amygdala, hippocampus or medial prefrontal cortex, but more interestingly we noticed that its expression was often higher (up to 100 fold) than that of MOR (*OPRM1*), KOR (*OPRK1*), DOR (*OPRD1*) and NOP (*OPRL1*) in the same region (Fig. 7a and Supplementary Fig. 7). These data were further confirmed by qPCR on human brain samples, where additional opioid centers such as dentate gyrus or locus coeruleus showed a similar high *ACKR3* expression (Fig. 7b).

Considering the expression of ACKR3 in the same regions of the CNS as the classical opioid receptors and its ability to efficiently internalize opioid peptides without inducing down-stream G protein mediated signaling, we wondered whether ACKR3 might influence classical opioid receptor signaling by regulating the availability of their ligands. To validate this hypothesis and the inability of ACKR3 to trigger signaling in a more physiological context, we used small molecule neural precursor cells (smNPCs)[53], that endogenously express ACKR3 but no classical opioid receptors (Fig. 7c). We confirmed that just like U87-ACKR3 cells, smNPCs express higher proportion of ACKR3 intracellularly compared to the cell surface (Fig. 7d) and that they are able to accumulate labeled dynorphin A (1–13)

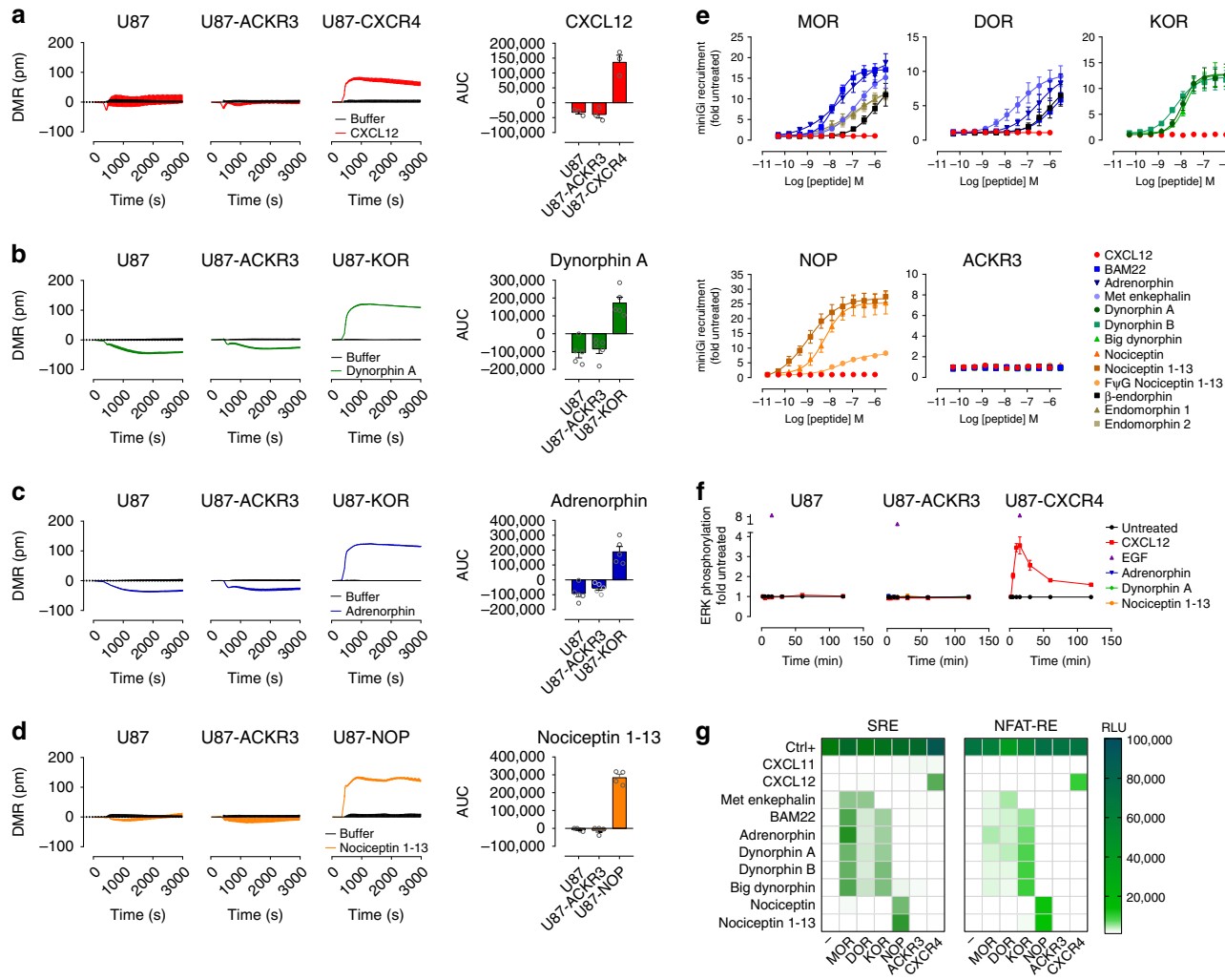

**Fig. 5 Absence of ACKR3 signaling in response to opioid and chemokine ligands. a–d** DMR profiles of U87 cells expressing (or not) ACKR3, CXCR4 or classical opioid receptors (KOR and NOP) stimulated by the chemokine CXCL12 (200 nM) (**a**) or opioid peptides (500 nM) dynorphin A (**b**), adrenorphin (**c**) and nociceptin 1–13 (**d**). Left panels: representative DMR profiles determined over 3000 s of three to five independent experiments. Right panel: area under the curve (AUC) ± S.E.M. of three to five independent experiments ($n = 3$ to 5). **e** Comparison of mini Gi recruitment to ACKR3 and classical opioid receptors (MOR, DOR, KOR, or NOP) in response to CXCL12 and opioid peptides monitored in U87 cells. The corresponding $EC_{50}$ and Emax values are summarized in Supplementary Table 7. **f** Kinetic analysis of ERK1/2 phosphorylation in U87 cells stably expressing ACKR3 or CXCR4 stimulated by CXCL12 and opioid peptides. EGF was used as positive control. **g** Comparison of activation of SRE (ERK1/2) and NFAT-RE ($Ca^{2+}$) signaling cascades (green color scale, relative light units, RLU) in U87 cells (-) or U87 expressing ACKR3, CXCR4 or classical opioid receptors (MOR, DOR, KOR, or NOP) in response to chemokines CXCL12 and CXCL11 (200 nM), opioid peptides (500 nM) or positive controls (30 nM PMA, 10% FBS for SRE and 30 nM PMA, 1 μM ionomycin, 10% FBS for NFAT-RE). Results represent the mean ± S.E.M. (**e–f**) or the mean (**g**) of three to five independent experiments ($n = 3$–5). Source data are provided as a Source Data file.

(Fig. 7e) without activating the ERK signaling pathway (Fig. 7f). In line with U87 cell results, this uptake was also associated with a decrease of the extracellular dynorphin A concentration and consequently its ability to signal through its corresponding classical opioid receptor (Fig. 7g).

To ultimately confirm the scavenging function of ACKR3 for opioid peptides, we monitored ex vivo the inhibition of spontaneous neuronal firing in slices of rat locus coeruleus, one of the brain regions where ACKR3 is found together with KOR and MOR (Fig. 7b)[54]. Treatment with dynorphin A led to a concentration-dependent inhibition of firing with total inhibition obtained with 1 μM (Fig. 7h). However, the same concentration of dynorphin A did not lead to a change in neuronal firing rate when pretreated with naloxone, indicating that the dynorphin A-induced inhibition of the firing can be attributed to a classical opioid receptor. Treatment with LIH383 at concentration as high as 3 μM however

did not lead to significant inhibition of neuronal firing, further confirming the inability of ACKR3 to trigger classical G protein signaling in this region of the CNS (Fig. 7h). Interestingly, pretreatment of locus coeruleus neurons with LIH383 (1 or 3 μM) to selectively block the scavenging capacity of ACKR3 resulted in an improved potency of dynorphin A towards its classical receptors (Fig. 7i). This observation is in line with our in vitro data and suggests that, in a physiological environment and at endogenous receptor abundance, ACKR3 exerts a scavenging function to regulate opioid peptide availability and thereby fine tune the signaling through their classical opioid receptors.

## Discussion

More than 40 years after the identification of the classical opioid receptors MOR, DOR and KOR and 20 years after the

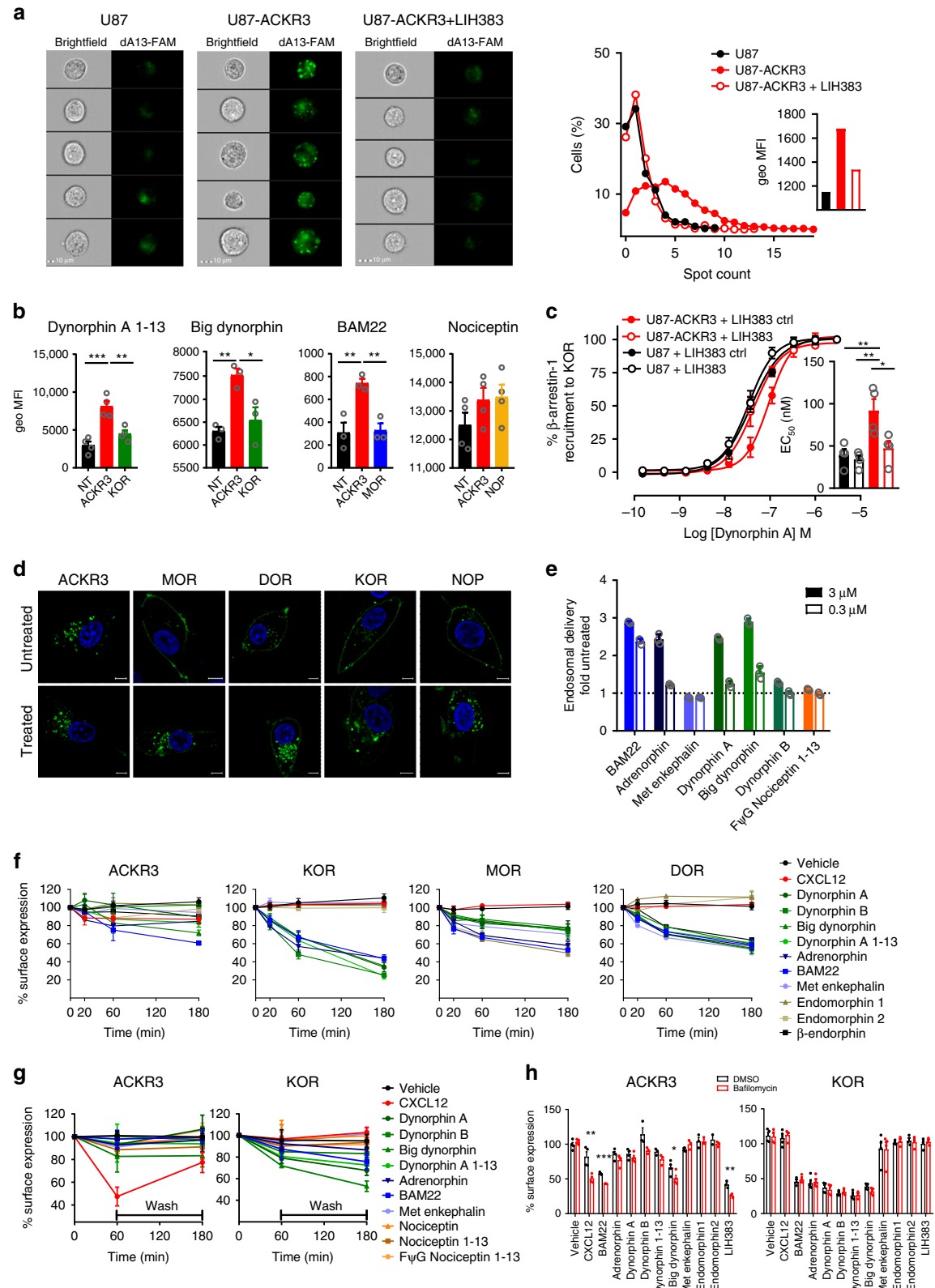

deorphanization of NOP, this study provides strong evidence that ACKR3 is an additional and broad spectrum opioid receptor. Besides sharing many endogenous ligands, ACKR3 and the classical opioid receptors are all expressed in different important centers of opioid activity in the CNS. Our study shows however, that ACKR3 is an atypical opioid receptor and does not induce

classical G protein signaling upon opioid peptide binding but rather internalizes these peptides in order to regulate their availability for classical opioid receptors.

ACKR3 was deorphanized as a chemokine receptor in 2006 based on its ability to bind CXCL12 and CXCL11. Our study demonstrates that in contrast to this rather narrow range of

**Fig. 6 Uptake of opioid peptides and atypical localization and trafficking of ACKR3. a** Uptake of dynorphin A (1–13) by ACKR3-expressing cells visualized by imaging flow cytometry. Left panels: U87, U87-ACKR3 or U87-ACKR3 cells pretreated with LIH383 (3 μM) stimulated with (FAM)-labeled dynorphin A (1–13) (250 nM, dA13-FAM, green channel). Five representative cells per condition are shown. Scale bar: 10 μm. Right panel: Percentage of cells with a given number of distinguishable vesicle-like structures (spots) and the geometrical mean fluorescence intensity (MFI, green channel) (inset). Data are representative of three independent experiments. **b** Uptake of opioid peptides (dynorphin A (1–13)-FAM (250 nM), big dynorphin-Cy5 (400 nM), BAM22-Cy5 (400 nM) or nociceptin-FAM (1 μM)) by U87 cells (NT) or U87 cells transfected with ACKR3 or classical opioid receptors analyzed by imaging flow cytometry as described in **a**. **c** ACKR3-mediated depletion of extracellular dynorphin A. U87 or U87-ACKR3 cells pretreated with LIH383 (400 nM) or LIH383ctrl were incubated with dynorphin A. Cell supernatant was added on U87 cells expressing KOR-SmBiT and LgBiT-beta-arrestin-1. (inset): $EC_{50}$ values. **d** Cellular localization of ACKR3 and classical opioid receptors fused to Neongreen fluorescent protein (green) stimulated or not by opioid peptides (1 μM) monitored by fluorescent confocal microscopy. Nuclear DNA was Hoechst-stained (blue). Pictures are representative of 10 acquired images from two independent experiments. Scale bar: 5 μm. **e** Ligand-induced receptor-arrestin delivery to endosomes monitored by β-galactosidase complementation assay in U2OS cells stably expressing ACKR3. Results are expressed as mean ± SD of three technical replicates. **f–h** Kinetics of ligand-induced internalization of ACKR3 and classical opioid receptors (**f** and **h**, HiBiT technology) and (**g**, flow cytometry). **f** U87 cells expressing N-terminally HiBiT-tagged receptors were stimulated with opioid peptides (1 μM) or CXCL12 (300 nM) for indicated times. Remaining membrane receptors were quantified with soluble LgBiT protein. **g** ACKR3 and KOR internalization and recycling in U87 cells after ligand stimulation (1 μM for opioid peptides, 300 nM for CXCL12) followed by acid wash, monitored by flow cytometry. **h** Effect of bafilomycin A1 (1.5 μM) on endosomal trafficking/cycling of ACKR3 and KOR following ligand stimulation (1 μM) monitored by HiBiT technology. If not otherwise indicated, results are presented as mean ± S.E.M of three to five independent experiments (*n* = 3 to 5). *p < 0.05, **p < 0.01, ***p < 0.001 by one-way ANOVA with Bonferroni correction (**b**), by two-way ANOVA: interaction between cell line and LIH383 treatment with Tukey's post hoc test (**c**), and by two-tailed unpaired *t*-test (**h**). Source data and statistical analysis parameters are provided as Source Data file.

chemokine ligands, ACKR3 is a highly promiscuous opioid receptor, binding and scavenging peptides belonging to different subfamilies, primarily enkephalins, and dynorphins, with potencies similar to those of classical opioid receptors. The function and signaling capacity of ACKR3 have since long been a matter of debate[24,26,47]. Our study, by using a range of different assay platforms—monitoring various early and late signaling steps typical of GPCRs—provides strong evidence that ACKR3, in contrast to classical opioid receptors and CXCR4, is not able to trigger detectable ERK activation or any other canonical G protein signaling in response to opioid peptides or chemokines, corroborating its atypical silent behavior. Moreover, our study shows that ACKR3 is not only capable of binding active opioid peptides but also their precursors, such as big dynorphin, as well as inactive N-terminally truncated peptides, such as dynorphin A 2–17, suggesting that it acts as an important multilevel rheostat of the opioid system. Although it is difficult to judge on the physiological relevance of these interactions that only occur at relatively high concentrations, it should be noted that certain opioid peptides, including dynorphin have been shown to reach micromolar local concentrations in certain areas of the CNS[55,56]. Indeed, in addition to the core-dependent inactivation of opioid peptides by broad-range proteolytic enzymes cleaving bonds within the N-terminal YGGF sequence[57–59], scavenging by ACKR3 provides an alternative way to fine tune the opioid system by specific address-dependent regulation of opioid peptide availability.

The identification of ACKR3 as a major regulator of the opioid system opens additional therapeutic opportunities. Indeed, drugs targeting the opioid system remain among the most widely prescribed analgesics for severe pain but their use frequently leads to tolerance, dependence or depression. There is thus an urgent need to find means to modulate the opioid system by drugs with alternative mechanisms of action and improved safety profiles. Our results showed that none of the small drugs targeting the classical opioid receptors such as morphine, fentanyl or naloxone activated or inhibited ACKR3. Interestingly, blocking ACKR3 with LIH383 positively impacted on the availability and signaling of opioid peptides through classical receptors in a rat ex vivo model, providing an original and indirect alternative to modulate the system. This concept is endorsed by previous in vivo results showing that mice treated with CCX771, a small ACKR3

modulator, exhibit anxiolytic-like behavior[34]. However, our results clearly point towards an arrestin-dependent opioid peptide scavenging function of ACKR3 in the CNS, rather than signal transduction, which is supported by the recent reconsideration of arrestins as having merely a regulatory role in GPCR signaling[44,60,61].

Besides ACKR3, other receptors with pharmacological properties incompatible with the classical receptors MOR, DOR, and KOR were proposed to bind opioid peptides. For instance, the N-methyl-D-aspartate (NMDA) receptor and the bradykinin receptor were reported to bind to big dynorphin and dynorphin A, respectively[62–64]. Other opioid activities were attributed either to the non GPCR Zeta (ζ) opioid receptor that binds met-enkephalin with high affinity, to the undefined opioid receptors Epsilon (ε) and Lamba (λ) or to other unknown receptors[65–67]. In the light of the present study, it is tempting to speculate that some of these activities may be attributed to the scavenging function of ACKR3[68,69]. Recently, the orphan receptor MRGPRX2 (Mas-related GPCR X2 or MrgX2) was also proposed as an AOR[55,70,71]. It is expressed on mast cells and small diameter neurons in dorsal root and trigeminal ganglia and induces classical G protein signaling in response to micromolar concentrations of pro-dynorphin-derived peptides and well-known synthetic opioid agonists. Its designation as an AOR stems from its restrained selectivity for dynorphin peptides, its unique preference for dextromorphinans and dextrobenzomorphans and its insensitivity to classical opioid receptor antagonists[55]. ACKR3, by analogy to its classification in the ACKRs subfamily, owes its designation as atypical to the characteristics that distinguish it from any other opioid peptide receptor. These include its inability to induce G protein signaling in response to ligand stimulation, its continuous recycling after opioid peptide binding and efficient ligand depletion from extracellular space, but also its broad-spectrum selectivity for opioid peptides, in the nanomolar range, and its unresponsiveness to alkaloid opioids and synthetic opioid drugs.

Similarly to the classical opioid receptors, ACKR3 is highly conserved among species and through the evolution indicating important functions. A detailed comparative sequence analysis revealed that ACKR3 does not harbor the residues delimiting the orthosteric morphinate binding pocket[72] of classical opioid receptors but does retain residues that are highly conserved in chemokine receptors[73] (Supplementary Fig. 3a and b). However, although structurally and genetically linked to the chemokine

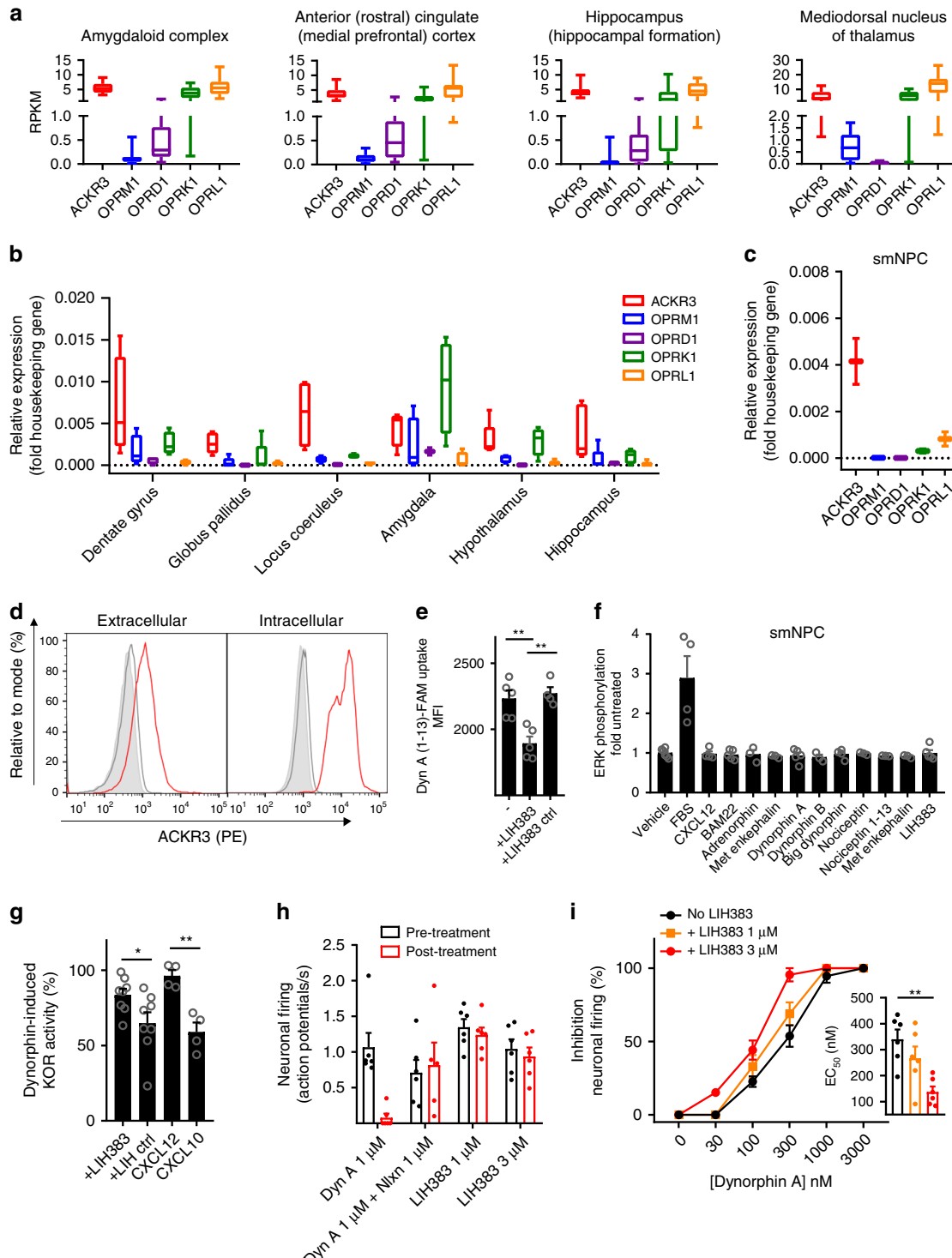

receptor family[73–76], phylogenetic analysis does not associate ACKR3 directly with the classical or ACKRs, but rather places it on a separate branch in between the chemokine and the opioid receptors (Supplementary Fig. 3c). Noteworthy, sequence analysis revealed the conservation of a four-residue loop within the N terminus of KOR ($C_{16}$APSAC$_{21}$) and ACKR3 ($C_{21}$NSSDC$_{26}$)[35,77]. Considering that few other GPCRs bear a similar loop, one might speculate that this loop derives from a common ancestor serving a similar purpose for both receptors.

Besides the brain, ACKR3 is abundantly expressed in numerous other tissues, including the adrenal glands where a link

between ACKR3 and BAM22 has previously been suggested[34]. The interplay between ACKR3 and opioids may therefore apply to other physiological systems and in particular the immune system, as subsets of lymphocytes, monocytes, or macrophages, proposed to express ACKR3, have also been shown to secrete and respond to opioid peptides[78–82]. The results of this study further strengthen the proposed crosstalk between the chemokine and the opioid systems but, by identifying a common receptor directly used by ligands of the two systems, extend it far beyond the concept of simple allosteric modulation through heterodimer formation[11,83,84]. How this dual ligand scavenging function

**Fig. 7 ACKR3-mediated regulation of opioid peptide availability for classical receptors. a–c** Relative gene expression of ACKR3 and classical opioid receptors in smNPCs and different brain regions corresponding to centers important for opioid peptide activity. Box plots encompass 25th to 75th percentile with median as central mark and whiskers extending to most extreme data points. **a** RNA-Seq RPKM (reads per kilobase per million) values from the open-source brainspan.org database ($n = 16$ for amygdaloid complex and mediodorsal nucleus of thalamus, $n = 15$ for hippocampus and $n = 17$ for anterior cingulate cortex). **b, c** mRNA expression determined by qPCR on five human adult brains (**b**) or two preparations of smNPCs (**c**) normalized to the arithmetic mean of PPIA and GAPDH as housekeeping genes. **d** Extracellular and intracellular expression of ACKR3 monitored by flow cytometry using ACKR3-specific mAb antibody (11G8, red) or a matched isotype control (MG1-45, gray) and a PE-conjugated secondary antibody in comparison to unstained cells (filled light gray), representative of two experiments with 10,000 gated cells. **e** Uptake of 250 nM (FAM)-labeled dynorphin A (1–13) by smNPCs pretreated with LIH383 or LIH383ctrl (3 μM) analyzed by imaging flow cytometry. **f** SRE (ERK1/2) signaling cascade activation in smNPC in response to various ligands (500 nM) or 10% FBS as positive control. **g** ACKR3-mediated depletion of extracellular dynorphin A. SmNPCs, pretreated with LIH383, LIHctrl (1.5 μM), CXCL12 or CXCL10 (300 nM), were incubated with dynorphin A (3 μM) and the activity of dynorphin A remaining in the cell supernatants was probed on U87 cells expressing SmBiT-tagged KOR and LgBiT-tagged mini Gi. Representative 30x supernatant dilution is shown. **h, i** Ex vivo rat locus coeruleus inhibition of neuronal firing induced by LIH383 alone, dynorphin A in the absence or presence of naloxone (**h**) or increasing concentrations of dynorphin A in the presence or absence of LIH383 (**i**) (inset): $EC_{50}$ values. For **e–i**, data are presented as mean ± S.E.M of independent experiments ($n = 3$ to 5 except in **g** for LIH and LIHctrl ($n = 8$)) and all animal experiments (**h** and **i**) ($n = 6$)). *$p < 0.05$, **$p < 0.01$ by one-way ANOVA with Bonferroni correction (**e** and **g**) and Kruskal–Wallis with two-sided Dunn's test (**i**). Source data and statistical analysis parameters are provided as Source Data file.

---

impacts on each other remains to be elucidated. Moreover, although the results of the present study add another piece to the growing body of evidence that ACKR3 is unable to trigger G protein signaling, it cannot be excluded that, in certain cell types or cellular contexts, ACKR3 could induce G protein-dependent or -independent signaling[27]. Finally, in analogy to its regulatory function within the chemokine system (when co-expressed with CXCR4) the ability of ACKR3 to heterodimerize with classical opioid receptors and to modulate their signaling properties remain to be investigated[85].

The unique dual chemokine-opioid peptide scavenging activity in the nanomolar range of ACKR3 represents a rare example of functional promiscuity among GPCRs (Fig. 1h). So far, few other receptor–ligand dualities have been described such as those between dopamine and noradrenergic receptors[86] and many such reports remain to be independently confirmed. This cross reactivity may stem from a common ancestor, co-evolution or the high degree of similarity between the opioid core and the N-terminal sequences of the ACKR3-binding chemokines.

In conclusion, the identification of ACKR3 as negative regulator of opioid peptide function adds another level of complexity and fine-tuning to the opioid system but also opens additional therapeutic opportunities. Considering its ligand-binding properties, lack of G protein signaling, and scavenging functions, we propose ACKR3 as a promiscuous, AOR and an opioid scavenger receptor.

## Methods

**Peptides and chemokines.** Non-labeled chemokines CXCL12, CXCL11, and vCCL2 were purchased from PeproTech. Alexa Fluor 647-labeled CXCL12 (CXCL12-AF647) was purchased from Almac. The opioid peptide library and all opioid peptides as well as FAM-labeled Dynorphin A (1–13) and FAM-labeled Nociceptin were acquired from Phoenix Pharmaceuticals. BAM22 and Big Dynorphin labeled with Cy5 were generated using an Amersham QuickStain Cy5 kit for proteins according to manufacturer's protocol. Adrenorphin-derived peptides were synthesized by JPT. These peptides contain a free amine at the N terminus and an amide group at the C terminus to avoid additional negative charge. Besides Levallorphan, which was purchased from Sigma, all non-peptide opioids were obtained from Tocris.

**Cell culture.** U87 cells derived from human brain glioblastoma were obtained through the NIH AIDS Reagent Program from Dr. Deng and Dr. Littman[87,88] and grown in Dulbecco's modified Eagle medium (DMEM) supplemented with 15% fetal bovine serum and penicillin/streptomycin (100 Units per ml and 100 μg per ml). U87-ACKR3 and U87-CXCR4 cells[36] were maintained under puromycin selective pressure (1 μg per ml). HEK293T and CHO-K1 (ATCC) cells were grown in DMEM supplemented with 10% fetal bovine serum and penicillin/streptomycin (100 Units per ml and 100 μg per ml). smNPC cells (small molecule neural precursor cells)[53] derived from a healthy donor (C1-1), whose informed consent was obtained[89], were grown on Geltrex™-coated surface in N2B27 medium

supplemented with 0.5 μM Purmorphamine, 3 μM CHIR 99021 and 150 μM ascorbic acid. N2B27 medium consisted of DMEM/F12 and NeuroBasal medium 50:50 with 0.5% $N_2$ supplement, 1% B27 supplement lacking vitamin A, 1% GlutaMAX and 1% penicilin/streptomycin. Medium was renewed every other day.

**Binding competition assays.** U87-ACKR3 cells were distributed into 96-well plates ($1.5 \times 10^5$ cells per well) and incubated with a mixture of 5 nM CXCL12-AF647 and unlabeled chemokines or opioid peptides at indicated concentrations for 90 min on ice, then washed twice with FACS buffer (PBS, 1% BSA, 0.1% $NaN_3$) at 4 °C. Dead cells were excluded using Zombie Green viability dye (BioLegend). ACKR3-negative U87 cells were used to evaluate non-specific binding of CXCL12-AF647. 0% receptor binding of CXCL12-AF647 was defined as the signal obtained after addition of 1 μM of unlabeled CXCL12. The signal obtained for CXCL12-AF647 in the absence of unlabeled chemokines was used to define 100% binding. Ligand binding was quantified by mean fluorescence intensity on a BD FACS Fortessa cytometer (BD Biosciences) using FACS Diva 8.01 (BD Biosciences).

**Nanoluciferase complementation-based assays.** Ligand-induced β-arrestin recruitment to chemokine and opioid receptors was monitored by NanoLuc complementation assay (NanoBiT, Promega)[35,90]. In brief, $1.2 \times 10^6$ U87 cells ($5 \times 10^6$ for HEK293T and $4 \times 10^6$ for CHO-K1) were plated in 10-cm culture dishes and 48 h (24 h for HEK293T and CHO-K1 cells) later cotransfected with pNBe vectors encoding GPCRs C-terminally tagged with SmBiT and human β-arrestin-1 (arrestin-2) or β-arrestin-2 (arrestin-3) or mini G proteins (mG, engineered GTPase domains of Gα subunits,) N-terminally fused to LgBiT[91–93]. 48 h post-transfection cells were harvested, incubated 25 min at 37 °C with Nano-Glo Live Cell substrate diluted 200-fold and distributed into white 96-well plates ($5 \times 10^4$ cells per well). Ligand-induced, β-arrestin and mini G recruitment to GPCRs was evaluated with a Mithras LB940 luminometer (Berthold Technologies, running on MicroWin 2010 5.19 software (Mikrotek Laborsysteme)) for 20 min. For single dose screening experiments on all chemokine receptors, the results are represented as percentage of signal monitored with 100 nM of one known agonist chemokine listed in the IUPHAR repository of chemokine receptor ligands which was added as positive control (Supplementary Table 3). For concentration–response curves, the signal recorded with a saturating concentration of full agonist for each receptor was set as 100%. To evaluate the antagonist properties of ligands, full agonists of each receptor (50 nM BAM22 for MOR, 50 nM dynorphin A for KOR, 70 nM met-enkephalin for DOR, 70 nM nociceptin for NOP and 4 nM CXCL12 for ACKR3) were added after the 20-min incubation with the ligands. Signal from wells treated with full agonist only was defined as 0% inhibition and signals from wells treated with no agonist were used to set 100% inhibition.

For dynorphin A scavenging experiments with U87 cells, $1.5 \times 10^5$ U87 or U87-ACKR3 cells were distributed per well in a white 96-well plate. After 15-min incubation at 37 °C with 400 nM LIH383 or LIH383 control peptide (200 nM CXCL12 or CXCL10), dynorphin A was added at concentrations ranging from 0.15 nM to 3 μM and incubated for 25 min at 37 °C. $1.5 \times 10^4$ U87 cells, cotransfected 48 h before the experiment with SmBiT-tagged KOR and LgBiT-tagged β-arrestin-1 or mini Gi and pre-incubated for 25 min with Nano-Glo Live substrate were then added per well and signal was measured for 20 min. For dynorphin A scavenging experiments with smNPC, $2 \times 10^6$ smNPC were pretreated for 15 min with 1.5 μM LIH383 or LIH383 control peptide (300 nM CXCL12 or CXCL10) before 4-h incubation with 3 μM dynorphin A. Cells were centrifuged and the activity of the remaining dynorphin A in serially diluted supernatants was determined on U87 cells expressing SmBiT-tagged KOR and LgBiT-tagged mini Gi protein.

**Label-free dynamic mass redistribution (DMR) assay.** Dynamic mass redistribution (DMR) experiments were conducted using the Corning Epic (Corning) biosensor system[43,94–97]. In brief, $6 \times 10^5$ U87 cells were seeded in 6-cm dishes. 24 h later cells were transfected with pcDNA3.1-based expression plasmids coding for the respective chemokine (ACKR3, CXCR4) or opioid (KOR, NOP) receptors. 24 h after transfection $1 \times 10^4$ cells per well were transferred to a 384-well Epic biosensor plate and incubated overnight at 37 °C.

Cells were then washed twice with Hanks' balanced salt solution (HBSS) (Life Technologies) containing 20 mM HEPES (Life Technologies) and subsequently incubated in the DMR-reader for 1.5 h to achieve temperature equilibration (37 °C). Five minutes after equilibration of the baseline DMR traces, compounds were added to the biosensor plate. Alterations of ligand-induced DMR were monitored for at least 3000 s. For quantification, negative and positive areas under the curve (AUC) between 0 and 3000 s were used.

**HTRF-based ERK1/2 phosphorylation assays.** HTRF-based phospho-ERK1/2 (extracellular signal regulated kinases 1 and 2) and total-ERK1/2 assays were performed using phospho-ERK1/2 (Thr202/Tyr204) and total-ERK1/2 cellular kits (Cisbio International). In short, for quantification of phosphorylated and total ERK1/2 protein, U87 cells stably expressing (or not) ACKR3 or CXCR4 were seeded in 96-well poly-D-lysine (PDL)-coated microtiter plates (Sigma-Aldrich) at a density of $3.5 \times 10^4$ cells per well. After overnight incubation, cells were starved for 4 h at 37 °C in serum-free medium. Cells were then stimulated for the indicated time intervals with chemokine or opioid ligands. The supernatants were replaced with the lysis buffer provided and incubated 1.5 h. Lysates were transferred to white 384-well plates and incubated with pERK1/2-specific (2 h) or total-ERK1/2-specific (24 h) antibodies conjugated with $Eu^{3+}$-cryptate donor and d2 acceptor at recommended dilutions. HTRF was measured using the Mithras LB 940 multimode reader (Berthold Technologies) equipped with 320 nm excitation filter and 620 nm (donor) and 665 nm (acceptor) emission filters.

**Transcriptional nanoluciferase reporter assays.** Activation of the MAPK/ERK signaling pathway was evaluated using a serum response element (SRE) Nanoluciferase reporter assay. Activation of calcium-dependent signaling pathways was evaluated using a Nuclear Factor of Activated T-cell response element (NFAT-RE) Nanoluciferase reporter assay. For both assays, $1.2 \times 10^6$ U87 cells ($5 \times 10^6$ for HEK293T, $4 \times 10^6$ for CHO-K1, $5 \times 10^6$ for smNPC) were seeded in 10-cm dishes and 48 h (24 h for HEK293T and CHO-K1 cells) later cotransfected with the pNanoLuc/SRE or pNanoLuc/NFAT-RE vectors (Promega), containing the Nanoluciferase gene downstream of SRE or NFAT-RE, and pcDNA3.1 encoding the respective chemokine or opioid receptors. 24 h later, $2.5 \times 10^4$ cells/well ($1 \times 10^5$ for HEK293T cells, $7 \times 10^4$ for CHO-K1 cells and $2.5 \times 10^5$ for smNPCs) were seeded in a white 96-well plate. 24 h later, the medium was replaced by serum-free and phenol red-free DMEM (serum-free DMEM/F12 for smNPCs) and further incubated for 2 h. Opioid peptides (500 nM) and chemokines (200 nM) were then added to the cells and incubated for 6 h. 30 nM phorbol 12-myristate 13-acetate (PMA), 10% FBS or 30 nM PMA, 1uM ionomycin, 10% FBS were used as positive controls for SRE and NFAT-RE assays, respectively. Nano-Glo Live Cell substrate (Promega) was then added and luminescence was read over 20 min on a Mithras LB940 plate reader (Berthold Technologies).

**Visualization of fluorescently labeled opioid-peptide uptake.** Cells were distributed into 96-well plates ($2 \times 10^5$ cells per well in Opti-MEM for U87 and U87-ACKR3 and $3 \times 10^5$ cells per well in N2B27 medium for smNPCs). After 15-min incubation at 37 °C with LIH383 (3 μM) or Opti-MEM only, FAM-labeled dynorphin A (1–13) (250 nM), BAM22-Cy5 (400 nM), big dynorphin-Cy5 (400 nM) or nociception-FAM (1 μM) was added, incubated for 40 min at 37 °C and washed twice with FACS buffer. For comparison of labeled opioid peptide-uptake by ACKR3 or classical opioid receptors, $1.2 \times 10^6$ U87 cells were seeded in 10-cm dishes and transfected 48 h later with 4 μg pcDNA3.1 plasmid encoding ACKR3 or KOR, MOR or NOP. 48 h post-transfection, cells were harvested and treated as described above. Dead cells were excluded using Zombie NIR or Zombie Green viability dye (BioLegend, #423106, dilution 1:1000 or #423112, dilution 1:3000 respectively) for FAM-labeled peptides and Cy5-labeled peptides, respectively. Images of $1 \times 10^4$ in-focus living single cells were acquired with an ImageStream MKII imaging flow cytometer (Amnis, running on the INSPIRE Mark II software (EMD Millipore)) using ×40 magnification (×60 magnification for smNPCs). Samples were analyzed using Ideas6.2 software. The number of spots per cell was determined using a mask-based software wizard.

**Receptors detection and localization analysis.** Intracellular and surface ACKR3 levels were analyzed by flow cytometry using ACKR3-specific mAb (12.5 μg per ml (dilution 1:40), clone 11G8 (R&D Systems, catalog #MAB42273) or a matched isotype control (12.5 μl per ml, (dilution 1:40) clone MG1-45, BioLegend, #401402) and phycoerythrin–conjugated F(ab')₂ fragment anti-mouse IgG (dilution 1:300, Jackson ImmunoResearch, #115-116-146). Dead cells were excluded using the Zombie NIR fixable viability dye (BioLegend, dilution 1:2000, catalog #423106). For intracellular staining, cells were treated with the BD Cytofix/Cytoperm Fixation/Permeabilization solution kit (BD Biosciences, catalog #554714) according to

manufacturer recommendations. Fluorescence intensity was quantified on a Novocyte Quanteon flow cytometer (ACEA Biosciences) using NovoExpress 1.4.1 (ACEA Biosciences) and samples were analyzed/processed using FlowJo 10.6.1.

For fluorescence imaging, $3 \times 10^5$ U87 cells/well of a 6-well plate were seeded and 48 h later transfected with 0.4 μg plasmid encoding ACKR3 or the opioid receptors C-terminally tagged with mNeonGreen. 24 h later, $1 \times 10^5$ cells were reseeded on 8-well chamber slides (μ-Slide 8 well, ibidi) and grown overnight. Cells were then incubated 90 min in the presence or absence of opioid peptides (1 μM) (dynorphin A for ACKR3, BAM22 for MOR, met-enkephalin for DOR, dynorphin A for KOR and nociceptin 1–13 for NOP) washed with PBS and fixed with 3.5% (w/v) paraformaldehyde for 20 min at room temperature. After three washes with PBS, nuclear staining was performed with Hoechst 33342 (1:1000) for 15 min at room temperature. Cells were again washed three times and imaged on a Zeiss LSM880 confocal microscope using a 63x oil-immersion objective using Zen Black 2.3 SP1 software (Zeiss). Representative cells from 10 image acquisitions of two independent experiments are shown.

**Ligand-induced receptor delivery to endosomes.** Opioid-peptide-induced receptor-arrestin complex delivery to endosomes was monitored by β-galactosidase complementation using a PathHunter eXpress ACKR3 activated GPCR internalization assay (DiscoverX). In brief, U2OS cells stably expressing ACKR3, β-arrestin-2 fused to the enzyme acceptor of β-galactosidase and an endosome marker fused to the β-galactosidase ProLink donor peptide were seeded 24 h before the experiment in a 96-well plate at a density of $1 \times 10^4$ cells per well. Opioid peptides (3 μM and 300 nM) were then added and after 4-h incubation at 37 °C, luminescent signal was generated through addition of 55 μl β-galactosidase substrate (PathHunter Detection reagent). After 1-h incubation at room temperature, chemiluminescent signal was measured on a Mithras LB940 plate reader (Berthold Technologies).

**Ligand-induced changes in receptor cell surface levels.** Determination of receptor surface expression level by NanoLuc complementation assay was performed using the Nano-Glo HiBiT extracellular detection system (Promega) according to manufacturer's protocol. In brief, $1.2 \times 10^6$ U87 cells were seeded on a 10-cm dish and 48 h later transfected with 100 ng plasmid encoding ACKR3 or the respective opioid receptors N-terminally tagged with HiBiT, a small part of the Nanoluciferase with high affinity towards LgBiT. 48 h later, $5 \times 10^4$ cells per well were seeded in 96-well plates and stimulated for the indicated time with CXCL12 (300 nM) or opioid peptides (1 μM) at 37 °C. Cells were then incubated with HiBiT extracellular reagent, consisting of Nanoluciferase extracellular substrate and LgBiT protein in HiBiT buffer. Light emission from complementation of LgBiT protein with remaining surface receptor-fused HiBiT was determined on a Mithras LB940 plate reader (Berthold Technologies). Signal was normalized to the measurement recorded at $t = 1$ min. Noteworthy the impact of nociceptin and derivatives could not be determined in this assay due to significant LgBiT protein cross-complementation by nociceptin. Where indicated, cells were treated with bafilomycin A1 (1.5 μM in 0.15% DMSO) (Santa Cruz Biotechnology) or 0.15% DMSO prior ligand stimulation (45 min) and during ligand stimulation (180 min).

For determination of receptor surface expression levels by flow cytometry, U87-ACKR3 or U87-KOR cells were stimulated with opioid peptides (1 μM) or CXCL12 (300 nM) for 60 min at 37 °C. The remaining surface-bound ligands were then removed by a brief wash with 150 mM NaCl, 50 mM glycine, pH 3 and twice with FACS buffer. Where indicated, cells were incubated for additional 120 min to allow surface receptor recovery. Cell surface levels of ACKR3 or KOR were then measured by flow cytometry at a saturating concentration (12.5 μg per ml, dilution 1:40) of receptor-specific mAb (clones 11G8 and 387301, R&D Systems, catalog #MAB42273 or #MAB3895, respectively) and a secondary phycoerythrin–conjugated F(ab')₂ fragment anti-mouse IgG (Jackson ImmunoResearch, cat. #115-116-146, dilution 1:300). Dead cells were excluded using the Zombie NIR fixable viability dye (BioLegend, catalog #423106, dilution 1:2000). Mean fluorescence intensity was quantified on a Novocyte Quanteon flow cytometer (ACEA Biosciences) using NovoExpress 1.4.1 (ACEA Biosciences).

**Ex vivo rat neuron firing rate.** Adult male Wistar rats (6–8-week old) were housed at room temperature in groups of three or four with a 12:12 h light-dark cycle. All animals had access to ad libitum food and water. All procedures were carried out in accordance with guidelines of the European Communities Council Directive of 24 November 1986 (86⁄609⁄EE) and were accepted by the Ethics Committee for Animal Use of the University of Liège (protocol 2061). All efforts were made to minimize animal suffering.

The methods used for brain slice preparation and recording procedures, were as previously described[98]. Rats were anaesthetized with chloral hydrate (400 mg per kg, i.p.) and placed under a cap with oxygenated air (95% O₂, 5% CO₂) 2 min prior to decapitation. After decapitation, the brain was rapidly removed and placed in ice cold (~2 °C) oxygenated artificial cerebrospinal fluid (aCSF) of the following composition: NaCl 130 mM, KCl 3.5 mM, NaH₂PO₄ 1.25 mM, NaHCO₃ 24 mM, Glucose 10 mM, CaCl₂ 2 mM, MgSO₄ 1.25 mM. A block of tissue containing the pons was placed in a vibrating blade microtome (Vibratome 1000 Plus, Sectioning System) and a slice containing the locus coeruleus (LC) immediately rostral to the fourth ventricle and the VIIth nerve, used as anatomical landmarks, was cut

coronally (400 μm thick). The slice was placed on a nylon mesh in a recording chamber (volume: 0.5 ml) where it was superfused by oxygenated aCSF (34.0 ± 0.5 °C) at a rate of 2–3 ml per min. The LC was recognized as a translucent region during transillumination, lateral to the fourth ventricle. All experiments were performed in oxygenated aCSF with synaptic blockers consisting of 10 μM CNQX, 10 μM SR95531, 1 μM MK801, and 1 μM CGP55845, which block AMPA/Kainate, GABAA, NMDA, and GABAB receptors, respectively. This ensured that the spontaneous firing of the neurons was only due to its endogenous pace making.

Extracellular single cell recordings of LC neurons were performed with glass microelectrodes filled with aCSF (resistance 10–20 MΩ). Signals were passed through an impedance adapter and were amplified 1000x using a homemade amplifier. They were displayed on a Fluke Combiscope oscilloscope and fed to an analog–digital interface (CED 1401, Cambridge Electronic Design, Cambridge, UK) connected to a computer. Data were collected and analyzed with the Spike 2 software (Cambridge Electronic Design, Cambridge, UK). All recorded neurons had a firing rate of 0.5–3 Hz with a good regularity (coefficient of variation of the interspike interval was $0.13 \pm 0.01$, $n = 18$) and a cessation of firing during application of the α2-adrenergic receptor agonist clonidine (10–20 nM). The duration of the extracellularly recorded action potentials was 2–3 ms. Drugs and peptides were applied for at least 10 min. Neuron firing was recovered to initial rates after the treatment was stopped.

The mean firing rate over 1 min was calculated during each condition. Next, the inhibition of firing by the peptides and drugs used (LIH383 and dynorphin) was quantified as the % of total inhibition. For this purpose, we considered the mean firing rate during the last minute of each condition (control, LIH383 alone, LIH383 plus a given concentration of dynorphin). The $EC_{50}$ of dynorphin was obtained using the Hill equation ($E/E_{max}$ = [dynorphin]/$EC_{50}$ (dynorphin) + [dynorphin]). One aberrant value (in the LIH383 3 μM group: 559 nM, which was >2 SD away from the mean value for this group) was omitted.

**RNA extraction and quantitative PCR**. Post-mortem samples from six brain regions of five patients suffering lethal non-head trauma were collected within 2–10 h of death as previously reported[99]. Brain autopsies were performed after informed consent of the closest relatives. Informed consent was also given for the use of anonymized brain tissues and clinical and pathological information for research purposes. The study was approved by the Ethical Committee of the Faculty of Medicine, Chiang Mai University, Thailand (protocol 2012-038). Total RNA was extracted from biopsies using the AllPrep DNA/RNA mini kit (Qiagen) or from smNPC using RNeasy mini kit (Quiagen) and stored at −80 °C until cDNA synthesis. First-strand synthesis was performed in a two-step process. Initially samples were incubated with RNaseOUT (Invitrogen) at 65 °C for 5 min. The reverse transcription reaction was subsequently performed at 55 °C for 60 min using Superscript III RT (Invitrogen) and 2 μM dT20 primer (Eurogentec). Quantitative PCR was performed on a CFX96 thermal cycler (Bio-Rad) and analyzed with CFX Manager 3.1 (Bio-Rad). Thermal cycling was performed as follows: denaturation at 95 °C for 15 min, 40 cycles at 95 °C for 15 s, annealing at 72 °C for 30 s, elongation at 72 °C for 30 s and a final elongation for 10 min at 72 °C. For each primer (Supplementary Table 8) the specificity of amplification was verified by melting curve analysis and visualization of PCR products on an agarose gel with SYBR Safe (Invitrogen). Relative PCR quantification was performed using the comparative threshold cycle method[100] using the arithmetic mean of PPIA and GAPDH as stable housekeeping genes. Samples with Ct values greater than three standard deviations from the mean were excluded from further analysis.

**Brainbank database analysis of gene expression**. CNS gene expression data were extracted from Allen Institute, BrainSpan: Atlas of the Developing Human Brain[101] (http://www.brainspan.org/static/download.html, file: RNA-Seq Gencode v10 summarized to genes). The dataset contains RNA-Seq RPKM (reads per kilobase per million) values averaged to genes[102]. For detailed descriptions of sample preparation, tissue selection criteria and data normalization, see the technical white paper, Developmental Transcriptome (http://help.brain-map.org/display/devhumanbrain/Documentation). Depending on the brain region, gene expression data were extracted form 16–22 donors aged from 4 months to 40 years old. Prenatal samples of the database were excluded (pcw 8-37). Brain samples from male and female donors were equally represented.

**Phylogenetic tree and sequence alignments**. Phylogenetic tree was created using sequences and the tree generator from the GPCRs database GPCRdb (https://www.gpcrdb.org/). For each GPCR, its full sequence was used to generate the tree using neighbor-joining distance calculation method. Design and formatting of the tree was done with CLC main workbench 7.9.1 using radial representation.

**Data and statistical analysis**. Concentration-response curves were fitted to the four-parameter Hill equation using an iterative, least-squares method (GraphPad Prism version 8.0.1). All curves were fitted to data points generated from the mean of at least three independent experiments.

All statistical tests, i.e. $t$-test, ordinary one way- or two-way ANOVA, Kruskal–Wallis test and post hoc analysis were performed with GraphPad Prism

8.0.1. Sample size was chosen to allow sufficient statistical power. $P$-values are indicated as follows: $*p < 0.05$, $**p < 0.01$, $***p < 0.001$, $****p < 0.0001$.

**Reporting summary**. Further information on research design is available in the Nature Research Reporting Summary linked to this Article.

## Data availability

All numerical source data underlying Figs. 1–7 and Supplementary Figs. 1–6 are provided as a Source Data file. Data presented in Supplementary Fig. 3 were generated using the GPCR database (www.gpcrdb.org). CNS gene expression data were extracted from Allen Institute, BrainSpan: Atlas of the Developing Human Brain (http://www.brainspan.org/static/download.html, file: RNA-Seq Gencode v10 summarized to genes). All other data are available from the corresponding author upon reasonable request.

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

## Acknowledgements

This study was supported by the Luxembourg Institute of Health (LIH) MESR (grants 20160116 and 20170113), Luxembourg National Research Fund (FNR) (INTER/FWO "Nanokine" - grant 15/10358798), Pathfinder "LIH383" (PF18/13255064), F.R.S.-FNRS-Télévie (grants 7456814 and 7461515). M. Meyrath and M. Merz are the Luxembourg FNR PhD fellows (grants AFR-3004509 and PRIDE-11012546 "NextImmune"). T.B. was funded by the Deutsche Forschungsgemeinschaft (DFG, German Research Foundation) - 214362475/GRK1873/2. E.K. was supported by the German research foundation (DFG)-funded research unit FOR2372 with the grants KO 1582/10-1 and -2. JO and RK were supported by the Luxembourg FNR (MaMaSyn and PEARL) and the Pelican Foundation. J.T. was supported by the Luxembourg FNR (C16/BM/11342695 "MetCOEPs", and C12/BM/3985792 "Epi-Path"). The authors wish to thank Manuel Counson and Nadia Beaupain for their technical help, Oliver Hunewald for his support in database and phylogenetic analyses, Maria Konstantinou and the Luxembourg national cytometry platform for their help with ImageStream analysis, Julien Hanson for critical reading of the manuscript and Joanna Muz for her assistance for figure design.

## Author contributions

M.M., M.S., and A.C. designed the study, interpreted the results and wrote the manuscript. M.M. carried out most experiments. J.Z. performed and analyzed DMR and pERK experiments, with the help from T.B., K.S., and E.K. L.M. and V.S. designed, performed, and analyzed the electrophysiological experiments on rat locus coeruleus neurons. J.T. provided brain tissues which were used in qPCR experiments performed and analyzed by M.P.M. J.O., and R.K. provided smNPC and related cDNA. E.K., V.S., and M.O. contributed to the concept of the study and to critical discussions. All authors contributed to the writing and approved the final version of the manuscript.

## Competing interests

A patent application has been filed on Novel Selective ACKR3 Modulators And Uses Thereof (Applicant: Luxembourg Institute of Health; inventors: A.C., M.M., and M.S.; PCT Application number: PCT/EP2020/061981). The remaining authors declare no competing interests.
