## [Peer Review File · Nature Communications]

Reviewers' comments:

Reviewer #1 (Remarks to the Author):

In this paper the authors nicely demonstrate that CXCR7 can interact with opioid peptides and that this interaction may have physiological and pharmacological significance. These are interesting and potentially important findings that will be widely discussed and the paper is likely to be widely read. I have several comments --relatively minor--for improving the paper. I have one major comment related to the suggestion that this is the first '5th' opioid receptor.

MAJOR COMMENT:

In 2017 Lansu et al, reported in Nature Chem Biol 2017 May;13(5):529-536 that MRGPRX2 represents another atypical opioid receptor--it is activated by several of the opioid peptides tested in this paper and, notably, is activated by many morphinans (in distinction to CXCR7). Indeed, the title of that paper classifies MRGPRX2 as an atypical opioid receptor. Parenthetically others have also noted MRGPRX2 can interact with opioids (Biochemical Pharmacology

Volume 146, 15 December 2017, Pages 214-223: Scientific Reports volume 8, Article number: 11628 (2018)). The relevance of this interaction for opioid side-effects has recently been demonstrated (Anesthesiology. 2019 Jul;131(1):132-147.).

I'm a bit surprised the authors 'missed' this particular paper. CXCR7 represents (at least) the second 'atypical opioid receptor'.

I would recommend that the authors delete 'fifth opioid receptor' from the title and elsewhere (it would clearly be the 6th) and call it 'a new atypical opioid receptor' instead. Also, the ways in which CXCR7 differs from MRGPRX2 should be mentioned in the discussion (probably merits a paragraph as readers would be interested in this).

That being said, another opioid-like receptor is interesting and potentially important!

MINOR COMMENTS:

1. Rajagopal et al, were the first to demonstrate CXCR7 activates arrestin translocation and this needs to be prominently mentioned (Proc Natl Acad Sci U S A. 2010 Jan 12;107(2):628-32.). Again here as this is such a highly cited and visible paper (and represents the initial discovery) it is surprising that the authors 'missed' this paper.

2. Line 152 'weak' rather than 'faint'

3. Line 368 'outlander' is not the right word here

4. It is not entirely clear that for all of the concentration-response curves the fitted parameters are provided. For each figure in the legend the location of the fitted parameters in a table or supplementary table needs to be provided.

5. Where differences in EC50 values are calculated one normally would perform an F-test (rather than a T-test) and the various statistical parameters provided (don't see them).

Reviewer #2 (Remarks to the Author):

This study aims to demonstrate that ACKR3 is a fifth opioid receptor and that it acts as a master scavenger for opioid peptides. This is an interesting area of research, but ACKR3 has already been reported to be an opioid receptor, and the conclusions presented in the title and abstract are not robustly supported by the data shown.

Although the authors have added to the list of opioids that bind ACKR3, it is already known that ACKR3 can bind to several opioid peptides (Ikeda et al (2013) Cell 155:1323) so the identification of ACKR3 as an opioid receptor is not novel. I was also confused by the authors proposed designation for ACKR3 as the 'fifth' opioid receptor after MOR, DOR, KOR and NOP given that they cite work in the Discussion section describing the opioid receptor activity of the NMDA receptor, the bradykinin receptor and the zeta opioid receptor (references 57 to 62).

The authors' conclusion that ACKR3 is a 'master scavenger of opioid peptides' is based primarily on the ability of U87 cancer cells expressing ACKR3 to accumulate fluorescence when incubated with one fluorescently labelled opioid (dynorphin A (1-13)) (Figure 7A). Many heptahelical receptors will

internalise ligand when they become activated, so the observation that cells expressing ACKR3 can internalise dynorphin A (1-13) is not compelling evidence of a specialised scavenging function for ACKR3. Chemokine scavenging by ACKR3 and other ACKRs is characterised by rapid receptor recycling, continuous ligand internalisation, and progressive depletion of extracellular ligand. These phenomena have not been investigated, so a scavenging function for ACKR3 has not been convincingly demonstrated.

The data in Fig 7D-E are also not compelling. In the preceding section of the results, LIH383 is shown to be a subnanomolar ACKR3 agonist, but much higher concentrations are used in Fig 7D-E. In E, for example, LIH383 only shows activity at 3 micromolar and 1 micromolar is ineffective. Off-target effects would appear to be a more likely than effects on ACKR3. Also, the use of the word 'master' suggests that it does the same thing to many opioid peptides - only one opioid peptide has been examined in ligand uptake assays so even if the evidence of scavenging activity was more compelling it is not appropriate to claim, as they do in the Abstract, that ACKR3 is a 'scavenger receptor for a wide variety of opioid peptides'.

I was surprised by the term 'non-signaling' used in the Abstract and elsewhere when referring to ACKR3's interaction with opioids because it seems to me that the opioid-induced ACKR3-dependent recruitment of b-arrestin that is assayed throughout the study is evidence of ligand-induced signalling. Even if this were, for some reason, excluded as a form of signalling, the authors should still not use the term 'non-signaling' because they have only examined signalling in cell lines artificially expressing ACKR3 and provide no evidence that ACKR3 does not signal in normal cells expressing it in the brain. Opioid-induced activation of MAPK through ACKR3 has been reported in other cell types (Ikeda et al (2013) Cell 155:1323) and there is lots of published evidence that chemokine binding to ACKR3, in addition to driving b-arrestin recruitment, activates other signal transduction pathways and modulates signalling through CXCR4 expressed on the same cell. Thus, it is incumbent on the authors to investigate opioid-induced signalling through, and indeed scavenging by, ACKR3 in physiological contexts in the brain before they draw any firm conclusions.

Thus, in short, the authors have a long way to go before they have proven that ACKR3 is a non-signalling master scavenger for opioid peptides.

In addition to these general comments, the following issues were noted and should be addressed by the authors.

1) There are some inaccuracies/omissions in the Introduction (Page 4)

Line 78: ACKRs have not recently emerged – they have been studied for many years since the Duffy antigen was shown to be chemokine binding protein.

Lines 80-81: Unlike other ACKRs, ACKR1 does not, to my knowledge, internalise and degrade ligands.

MIF is also reported to be an ACKR3 ligand and should be mentioned in the Introduction.

2) It would be helpful if there were more complete details included in the Figure Legends. In many cases, it was difficult to find all the information required to understand the Figures (e.g. chemokine concentrations (Fig 1); cell type used (Fig 1); cell ratios (Fig 7E)).

3) Positive controls are required to avoid false negatives in Figure 2 & S2.

4) From Fig S1A and Fig 1A it appears that μ -endorphin and big dynorphin might have some effect on ACKR3 and CXCR3, respectively. What statistical tests have been used to exclude these data as significant?

5) Page 7. Lines 132-4. A lot of ligand seems to be required to get full activation, so these data should be discussed in the context of their EC50s and likely physiological relevance.

6) It is not clear how Figure 3 informs the subsequent parts of the study and the conclusions (lines 178-180) are speculative. It should be included as Supplementary Figure and its description and discussion moved to the Discussion section. Others have done detailed phylogenetic analysis of chemokine receptors (lines 175-178) and this work should be cited (e.g. Nomiya et al (2011) *Dev Comp Immunol* 35:705).

7) Page 11: Lines 235-6: It would be helpful if suitable publications were cited to highlight those small molecules for ACKR3 modulation that have been developed (e.g. Wang et al (2018). *Front Pharmacol* 9:641).

8) The expression of ACKR3 has been examined in the brains of mice and humans in a variety of ways (e.g. Banisadr et al *J Neuroimm Pharmacol* 11:26, Shimizu et al *PLoS One* 6:e20680 and others). The authors should cite this work and should make it more explicit how their data in Fig 7B-C add to current knowledge.

9) The authors should use accepted format when referring to genes/transcripts vs proteins i.e. uppercase, lower case, italics etc.. It makes it easier to understand what is being examined.

10) In Figure 7D, instead of using untransfected U87 cells as controls, would it be better to use U87-KOR cells (with no b-arrestin reporter) to allow the authors to see if ACKR3 has more opioid scavenging activity than KOR?

11) Pages 23-24. The authors point out that datapoints have been removed from the analysis. The full dataset should be presented so that the reader can draw their own conclusions. As I understand it, t-tests are only appropriate test to use when there are two groups in an experiment so an alternative statistical test should be applied to the data in Figure 7D and S6 and the interpretation changed if required.

12) Lines 376-377. The work cited does not seem to refer to ACKR3-expressing lymphocytes, monocytes or macrophages, or to the secretion of opioid peptides. Some clarification is required.

13) Did the authors use mouse or human chemokines in their study?

Reviewers' comments:

We would like to thank the reviewers and the editorial team of Nature Communications for the time taken to assess our work. The constructive concerns that have been raised led to major improvements of the manuscript. The manuscript was modified according to the reviewers' requests and their concerns were addressed as explained below.

Reviewer #1 (Remarks to the Author):

In this paper the authors nicely demonstrate that CXCR7 can interact with opioid peptides and that this interaction may have physiological and pharmacological significance. These are interesting and potentially important findings that will be widely discussed and the paper is likely to be widely read. I have several comments --relatively minor--for improving the paper. I have one major comment related to the suggestion that this is the first '5th' opioid receptor.

→ Answer: *We thank reviewer 1 for his/her positive feedback and comments. The manuscript was modified according to the reviewers' requests and the concerns were addressed as explained.*

MAJOR COMMENT:

In 2017 Lansu et al, reported in Nature Chem Biol 2017 May;13(5):529-536 that MRGPRX2 represents another atypical opioid receptor-it is activated by several of the opioid peptides tested in this paper and, notably, is activated by many morphinans (in distinction to CXCR7). Indeed, the title of that paper classifies MRGPRX2 as an atypical opioid receptor. Parenthetically others have also noted MRGPRX2 can interact with opioids (Biochemical Pharmacology Volume 146, 15 December 2017, Pages 214-223; Scientific Reports volume 8, Article number: 11628 (2018)). The relevance of this interaction for opioid side-effects has recently been demonstrated (Anesthesiology. 2019 Jul;131(1):132-147.).

I'm a bit surprised the authors 'missed' this particular paper. CXCR7 represents (at least) the second 'atypical opioid receptor'. I would recommend that the authors delete 'fifth opioid receptor' from the title and elsewhere (it would clearly be the 6th) and call it 'a new atypical opioid receptor' instead. Also, the ways in which CXCR7 differs from MRGPRX2 should be mentioned in the discussion (probably merits a paragraph as readers would be interested in this).

That being said, another opioid-like receptor is interesting and potentially important!

→ Answer: *In light of this comment, the title of our manuscript was changed from "A fifth opioid receptor: ACKR3/CXCR7 acts as a master scavenger for opioid peptides" to "The atypical chemokine receptor ACKR3/CXCR7 is a broad-spectrum scavenger for opioid peptides".*

The studies describing MRGPRX2 as an opioid receptor are now discussed and referenced (see refs 55, 70 and 71). A paragraph explaining the differences in the "atypical" properties of MRGPRX2 versus ACKR3 was added in the discussion as suggested (see lines 391-403). This new paragraph compares the ligand selectivity and signaling vs scavenging properties of the two receptors.

MINOR COMMENTS:

1. Rajagopal et al, were the first to demonstrate CXCR7 activates arrestin translocation and this needs to be prominently mentioned (Proc Natl Acad Sci U S A. 2010 Jan 12;107(2):628-32.). Again here as this is such a highly cited and visible paper (and represents the initial discovery) it is surprising that the authors 'missed' this paper.

→ **Answer:** *Although the study of Rajagopal et al. 2010 (PNAS 12;107(2):628-32.) had already been cited in the initial version of the manuscript (see ref 51 now 26), the reference has now also been added to support the statement that ACKR3 activation leads to the recruitment of arrestin to the receptor (see line 85).*

2. Line 152 'weak' rather than 'faint'

→ **Answer:** *The text was modified as suggested (see line 148).*

3. Line 368 'outlander' is not the right word here

→ **Answer:** *Following the recommendation of the Reviewer 2, the related paragraph was moved from the results section to the discussion and was shortened (see lines 404 to 414). The sentence containing the word "outlander" was removed during this process.*

4. It is not entirely clear that for all of the concentration-response curves the fitted parameters are provided. For each figure in the legend the location of the fitted parameters in a table or supplementary table needs to be provided.

→ **Answer:** *All the data corresponding to the different fitted parameters are now presented as supplementary data (see Table 1 and supplementary Tables 2-6, 8 and 9) and the references to each table are included in each figure legend (see lines 739, 766, 782 and 794).*

5. Where differences in EC₅₀ values are calculated one normally would perform an F-test (rather than a T-test) and the various statistical parameters provided (don't see them).

→ **Answer:** *We performed an F-test for the comparison of the EC₅₀ values presented in Fig. 6c and supplementary Fig. 6. For the experiments conducted in rat brains (Fig. 7i), a nonparametric Kruskal Wallis was performed (see lines 838 and 868 of main manuscript and 138 of supplementary information). The corresponding parameters are now presented as supplementary data in an excel file.*

Reviewer #2 (Remarks to the Author):

This study aims to demonstrate that ACKR3 is a fifth opioid receptor and that it acts as a master scavenger for opioid peptides. This is an interesting area of research, but ACKR3 has already been reported to be an opioid receptor, and the conclusions presented in the title and abstract are not robustly supported by the data shown.

Although the authors have added to the list of opioids that bind ACKR3, it is already known that ACKR3 can bind to several opioid peptides (Ikeda et al (2013) Cell 155:1323) so the identification

of ACKR3 as an opioid receptor is not novel. I was also confused by the authors proposed designation for ACKR3 as the 'fifth' opioid receptor after MOR, DOR, KOR and NOP given that they cite work in the Discussion section describing the opioid receptor activity of the NMDA receptor, the bradykinin receptor and the zeta opioid receptor (references 57 to 62).

→ **Answer:** *We thank the reviewer for the time taken to assess our work and for his/her feedback and comments. The manuscript was modified and the concerns raised were addressed as explained below.*

Firstly, to accommodate the reviewers' comments, the designation of ACKR3 as a fifth opioid receptor was removed from the manuscript and the title was changed to "The atypical chemokine receptor ACKR3/CXCR7 is a broad-spectrum scavenger for opioid peptides". Indeed, our study demonstrates that ACKR3 is a broad-spectrum receptor for opioid peptides not only from the met-enkephalin family as reported by Ikeda et al. for BAM22 but also from the dynorphin and the nociceptin families. This broad-spectrum selectivity is atypical and unique among the opioid receptors. We also demonstrate that ACKR3, in contrast to other opioid receptors, is not able to induce G protein signalling in response to opioid peptide stimulation, is present in different cellular compartments and acts as a scavenger receptor regulating the availability of opioid peptides for classical signalling receptors. We propose thus a new role for ACKR3 as a negative regulator of opioid peptides. We indeed show in a rat ex-vivo model that blocking ACKR3 increased the availability and signalling induced by opioid peptides through classical receptors. Finally, this study proposes an original and indirect alternative of modulating the opioid system by drugs (ACKR3 modulators) with novel mechanisms of action and potentially improved safety profiles.

We do agree that other receptors, including NMDA receptor, bradykinin receptor and zeta opioid receptor, listed in the discussion, were also proposed to act as opioid receptors. However, their affinity, their restricted ligand selectivity, often limited to one peptide (e.g. met-enkephalin for zeta opioid receptor or dynorphin A and derivatives for bradykinin receptor) and their ability to signal through G proteins make them very different from ACKR3 that shows the unique properties described in the present study.

In addition to the existing paragraph describing the NMDA, bradykinin and the zeta opioid receptors (lines 385 to 389), a paragraph explaining the differences between ACKR3 and MrgX2, another receptor recently proposed as an "atypical opioid receptor" based on its unconventional ligand-binding profile (dynorphin and morphinan analogues) was included as suggested by the reviewer 1 (see lines 391-395). A paragraph on the atypical properties of ACKR3 in comparison to all these other proposed opioid receptors was added in the discussion (see lines 395-403).

The authors' conclusion that ACKR3 is a 'master scavenger of opioid peptides' is based primarily on the ability of U87 cancer cells expressing ACKR3 to accumulate fluorescence when incubated with one fluorescently labelled opioid (dynorphin A (1-13)) (Figure 7A). Many heptahelical receptors will internalise ligand when they become activated, so the observation that cells expressing ACKR3 can internalise dynorphin A (1-13) is not compelling evidence of a specialised scavenging function for ACKR3. Chemokine scavenging by ACKR3 and other ACKRs is characterised by rapid receptor

recycling, continuous ligand internalisation, and progressive depletion of extracellular ligand. These phenomena have not been investigated, so a scavenging function for ACKR3 has not been convincingly demonstrated.

→ **Answer:** *We understand the reviewer's criticism that the set of data illustrating the broad-spectrum scavenging function of ACKR3 was somewhat limited in the initial version of our manuscript. This aspect of our study was carefully addressed in the revised version of the manuscript.*

The scavenging function of ACKR3 is already well established for the chemokines CXCL12 and CXCL11 including in the brain (Naumann et al. (2010) PLoS One e9175 and Saaber et al (2019) Cell reports 1473-1488, ref. 50 and 47 in the manuscript). The hypothesis for a similar role for opioid peptides that is put forward in this study was first inferred from the inability of ACKR3 to induce intracellular signalling in response to opioid peptide stimulation. This scavenging function was illustrated using brain-derived cell line U87 and in a rat ex vivo model of locus coeruleus with endogenous expression of ACKR3 and classical opioid receptors. In the revised version of the manuscript, opioid peptide scavenging was further confirmed in the rat ex vivo model of locus coeruleus and was also confirmed in neural precursor cells (smNPC) (Fig. 7h and 7e-g).

As for the ACKR3 selectivity, already in the initial manuscript, supported by several complementary observations, including binding competition data and receptor activation (beta arrestin recruitment) data presented Figure 1b-g, Table1 and supplementary Figure 1) we clearly showed that ACKR3 is a broader spectrum receptor for opioid peptides than the 4 classical opioid receptors. The broad-spectrum scavenger function of ACKR3 is now shown in fluorescent ligand uptake experiments for an extended set of opioid peptides from different opioid families (BAM22, big dynorphin and nociceptin) and strengthened by additional internalization and endosome delivery results generated using different techniques (see below and Fig. 6b and 6e-h).

Additional experiments and controls were performed to address the question of the "recycling, continuous ligand internalisation, and progressive depletion of extracellular ligand" raised and to further illustrate the opioid peptide scavenging capacity of ACKR3:

A section (2.7, lines 262-304) supported by a new figure was added in the main text showing differences between ACKR3 and the classical opioid receptors in terms of opioid peptide uptake, internalisation/cycling profiles in response to opioid peptide stimulation (higher recycling rate for ACKR3 compared to classical opioid receptors) (Fig. 6f-g) and cellular localisation (mostly intracellular for ACKR3 in contrast to the cell surface presence of classical opioid receptors) (Fig. 6d). Furthermore, the revised manuscript includes data demonstrating that the stimulation by a large variety of opioid peptides induces the delivery of the ACKR3-arrestin complex to the endosomes, despite only minor changes of net surface levels of ACKR3, strongly pointing to a continuous recycling function of ACKR3. We also report data showing the ability of ACKR3 to efficiently deplete dynorphin A from the extracellular space using U87 cells (Fig. 6c) but also with neural precursor cells (smNPC) endogenously expressing ACKR3 (Fig. 7c-e, and 7g, lines 319-327).

Regarding Figure 7a, now Figure 6a, additional experiments were performed with cells expressing KOR, the classical opioid receptor for dynorphin A (1-13), in comparison to ACKR3-expressing cells to

show that although classical opioid receptors internalise opioid peptides, their uptake is far less efficient than for ACKR3. Indeed, the results show that although KOR has a higher affinity for dynorphin A 1-13 ($EC_{50} = 2.5 \text{ nM}$) than ACKR3 ($EC_{50} = 62 \text{ nM}$), the uptake of dynorphin-A 1-13 by the cells expressing ACKR3 is significantly higher than that observed with the cells expressing KOR (see lines 271-273 Fig. 6b). Similar results were obtained for big dynorphin, the precursor of dynorphin A and dynorphin B, BAM22 and nociceptin. For each of these experiments we compared the uptake by ACKR3 and the corresponding classical opioid receptor (KOR, MOR or NOP).

Finally, we included additional results showing that inhibition of neuron firing cannot be achieved by specific activation of ACKR3, but only through activation of classical opioid receptors, whereas neutralisation of ACKR3 scavenging capacity clearly demonstrates an improved potency of dynorphin A towards its classical receptors (see lines 330-339 and Fig. 7h-i).

The data in Fig 7D-E are also not compelling. In the preceding section of the results, LIH383 is shown to be a subnanomolar ACKR3 agonist, but much higher concentrations are used in Fig 7D-E. In E, for example, LIH383 only shows activity at 3 micromolar and 1 micromolar is ineffective. Off-target effects would appear to be a more likely than effects on ACKR3.

→ **Answer:** We agree with the reviewer that the micromolar concentrations (1 μM or 3 μM) of peptide LIH383 used in the ex vivo experiments may appear high at first sight. However, these experiments were performed on 400- μm thick rat brain slices, which were superfused with dynorphin-A or LIH383 peptides. It is known that in such conditions peptides usually appear less potent than on cell lines, in part because of the incomplete equilibration of their concentration within the slices but also because of their possible retention in the biological matrix or their degradation by peptidases in tissue slices. Similar observation can be made for dynorphin-A, which on cell lines activates KOR with EC_{50} ranging from 12.5 nM (beta-arrestin-1 assay) to 11.8 nM (mini Gi assay) while a concentration of 1 μM was needed for the complete inhibition of neuron firing in the slices, with an EC_{50} value of 340 nM, in the absence of LIH383. In these conditions, it is expectable that the concentrations of LIH383 necessary to observe an effect in brain slices would be higher than the concentration of dynorphin A peptide added.

In the revised manuscript, we included data (Fig. 7h) showing that treatment with dynorphin A led to a concentration-dependent inhibition of firing with total inhibition obtained with 1 μM , whereas treatment with the same concentration of dynorphin A lead to no change in neuronal firing rate when pre-treated with naloxone, an antagonist of KOR, DOR and MOR but not of ACKR3, indicating that the dynorphin A-induced inhibition of the firing can only be attributed to a classical opioid receptor and not to ACKR3. Moreover the new data included show that treatment with LIH383 at a concentration as high as 3 μM alone did not lead to significant inhibition of neuronal firing, further confirming the different behaviour of ACKR3 and pointing to its inability to trigger classical G protein signalling in the CNS and suggesting that the observed effects cannot be attributed to off-target effects. However, as presented in the initial version of the manuscript, pre-treatment with LIH383 (1 or 3 μM), to selectively block the ligand binding capacity of ACKR3, resulted in an improved potency

of dynorphin A towards its classical receptors corroborating the scavenging and negative regulatory function of ACKR3 for opioid peptides in a more physiological context.

Also, the use of the word ‘master’ suggests that it does the same thing to many opioid peptides - only one opioid peptide has been examined in ligand uptake assays so even if the evidence of scavenging activity was more compelling it is not appropriate to claim, as they do in the Abstract, that ACKR3 is a ‘scavenger receptor for a wide variety of opioid peptides’.

→ Answer: *The designation “master” scavenger was removed from the title and the abstract. In the revised version of the manuscript, we provide, in addition to the binding and arrestin data, new uptake (Fig. 6b-c, 7e and 7g), internalisation (Fig. 6f-h) and endosome delivery (Fig. 6e) data demonstrating that ACKR3 is a broad-spectrum scavenger for different opioid peptides from the enkephalin, dynorphin and nociceptin families. These additional results are presented in Figures 6 and 7, see lines 262-304, 319-327, 804-840 and 848-863.*

I was surprised by the term ‘non-signaling’ used in the Abstract and elsewhere when referring to ACKR3’s interaction with opioids because it seems to me that the opioid-induced ACKR3-dependent recruitment of b-arrestin that is assayed throughout the study is evidence of ligand-induced signalling. Even if this were, for some reason, excluded as a form of signalling, the authors should still not use the term ‘non-signaling’ because they have only examined signalling in cell lines artificially expressing ACKR3 and provide no evidence that ACKR3 does not signal in normal cells expressing it in the brain. Opioid-induced activation of MAPK through ACKR3 has been reported in other cell types (Ikeda et al (2013) Cell 155:1323) and there is lots of published evidence that chemokine binding to ACKR3, in addition to driving b-arrestin recruitment, activates other signal transduction pathways and modulates signalling through CXCR4 expressed on the same cell. Thus, it is incumbent on the authors to investigate opioid-induced signalling through, and indeed scavenging by, ACKR3 in physiological contexts in the brain before they draw any firm conclusions. Thus, in short, the authors have a long way to go before they have proven that ACKR3 is a non-signalling master scavenger for opioid peptides.

→ Answer: *We thank the reviewer for having highlighted this important issue related to the signalling capacity of ACKR3. We understand the reviewer’s concerns and we feel that some of the statements and observations from our study as well as from the study invoked by the reviewer (Ikeda et al (2013) Cell 155:1323), require clarifications.*

In the initial manuscript, we defined ACKR3 activation as recruitment of beta-arrestin to the receptor following ligand stimulation (see lines 85-86 and 115-116, now 83-84 and 111-112), with the aim to differentiate receptor activation from downstream signalling events, the triggering of which is highly debated for ACKR3. We take the reviewer’s point that beta-arrestin recruitment can be considered as a form of signalling. We also agree that the term “non-signalling” may be too general and not the most appropriate as for instance we cannot exclude that ACKR3 triggers activation of signalling pathways different from those activated downstream G protein. We therefore removed the denomination of ACKR3 as “non-signalling” receptor throughout the manuscript or specified where

necessary that it implies a lack of signalling through classical G-protein signalling pathways (see lines 336, 350 and 438).

Indeed, in the initial version of our study, we applied various technologies such as Dynamic Mass Redistribution (DMR) and monitored changes in several intracellular effectors related to signal transduction events typical to GPCRs (G proteins, calcium, ERK). We performed these experiments in various cell lines, including U87 cells that have a neural/brain origin, but also in HEK and CHO cells. No activation of these classical GPCR-related signalling pathways was observed following ACKR3 stimulation with its cognate chemokine CXCL12 or opioid peptides.

In the revised manuscript, to accommodate the reviewer's comment on the "artificial system" aspect of investigating signalling events in cell lines, we demonstrate that stimulation of ACKR3 in neuron precursor cells endogenously expressing ACKR3 does not lead to activation of ERK signalling pathways. These results are also further strengthened by additional controls in the ex-vivo experiments performed in rat brain slices in which we monitored the activity of LIH383 and opioid peptides on primary neurons and neural cells. Indeed, we included data (Fig. 7h) showing that specific activation of ACKR3 by LIH383 at concentrations as high as 3 μ M did not lead to significant inhibition of neuronal firing in rat brain slices. In contrast, treatment with dynorphin A led to a concentration-dependent inhibition of firing with total inhibition obtained with 1 μ M, whereas treatment with the same concentration of dynorphin A led to no change in neuronal firing rate in the presence of naloxone, an antagonist of KOR, DOR and MOR but not of ACKR3. This indicates that the dynorphin A-induced inhibition of the firing can only be attributed to a classical opioid receptor and not to ACKR3 (Fig. 7h). However, as presented in the initial version of the manuscript, pre-treatment with LIH383 (1 or 3 μ M), to selectively block the ligand binding capacity of ACKR3, resulted in an improved potency of dynorphin A towards its classical receptors corroborating the scavenging and negative regulatory function of ACKR3 for opioid peptides in a more physiological context. Altogether these data further confirm the different behaviour of ACKR3 and point to its inability to trigger classical G protein signalling in the CNS (see Fig. 7h-i and lines 328-341).

We also agree with the reviewer that several studies suggested that ACKR3 could trigger direct ERK/MAPK signalling. This possibility is currently highly debated and references to this were added (see lines 83 to 85 and lines 238 to 239 and refs 26-29 and refs 45-50). Regarding the study specifically invoked by the reviewer, (Ikeda et al. (2013) Cell 155:1323), the results reported reveal no direct ERK phosphorylation following the treatment of the H295R cells (adrenal gland derived cell line) with BAM22 alone (see below "Figures 5C, 5E and 6C" Ikeda et al. (2013)). The only effect that the authors could observe was when the cells were treated with a combination of BAM22 (activating ACKR3) and ACTH, the agonist of the melanocortin receptor 2 (MC2R) another receptor expressed by H295R cells. Their results demonstrated that BAM22 can enhance the effect of ACTH but they do not provide evidence that BAM22 can trigger ERK phosphorylation by itself (via ACKR3). The schematic representation of the opioid-glucocorticoid interactions with ACKR3 (see below "Figures 6A" Ikeda et al. (2013)) may lead to believe that ACKR3 stimulated by BAM22 triggers beta-arrestin recruitment and subsequent ERK phosphorylation but this claim is not supported by the experimental data (see "Figure 5E" Ikeda et al. (2013)) (see red marks).

Figure 5. CXCR7 mediates the enhanced circadian glucocorticoid oscillation induced by SCH (A) *In situ* hybridization with cRNA probes for *Cxcr7*, *Oprd1*, *Oprk1*, *Oprm1* and *Mrgprx1*. 2mo, 2-month-old; 4mo, 4-month-old; 6mo, 6-month-old. Scale bar, 500 μ m. (B) Effects of CXCR7-neutralizing antibody treatment on enhanced circadian glucocorticoid oscillation induced by SCH (ZT0; n = 4/group, ZT12; n = 7-16/group). (C) Effects of CXCR7 knockdown on enhanced Cortisol secretion by BAM22. (D) Effects of β -arrestin-1 and -2 knockdown on enhanced Cortisol secretion by BAM22. (E) Enhanced ACTH-induced ERK1/2 phosphorylation by BAM22 (3 μ M). (F) Effects of U0126 (1 μ M) treatment on the stimulatory action of BAM22.

Figure 6. Activation of CXCR7 by CCX771 is sufficient to induce anxiety-like behavior (A) Schematic of the opioid-CXCR7-glucocorticoid pathway (gray rectangle) linking adrenal tissue remodeling to behavior. (B) Activation of CXCR7 by CCX771 in a β -arrestin-2 recruitment assay. (C) CCX771 enhanced Cortisol secretion from H295R cells in the presence of ACTH (+; 300 nM, ++; 1 μ M). (D) CCX771 (100 nM) augmented corticosterone secretion in isolated adrenal slices from 6-month-old female mice in the presence of ACTH (n = 6-10/group). (E) CCX771 injection regimen. (F) Increased circadian amplitude of glucocorticoid secretion by CCX771 (n = 6-8/group). (G) Anxiety-like effects of CCX771 in the elevated plus maze test (n = 17-18/group). (H) Effects of a

Nevertheless, in order not to exclude possible alternative signalling pathways or indirect effects through for instance receptor dimerization, we added a paragraph at the end of the manuscript mentioning that although our study strongly suggests that ACKR3 does not mediate classical G protein signalling in response to opioid peptide binding, other forms of signalling by ACKR3 may occur (see lines 423 to 426). This paragraph also discusses that in analogy to the regulatory function of ACKR3 within the chemokine system (when co-expressed with CXCR4) the ability of ACKR3 to heterodimerize with classical opioid receptors and to modulate their signalling properties should be investigated (see lines 426-429).

We hope that we addressed the reviewer's questions and that the clarifications and the additional experiments conducted provide enough convincing arguments.

MINOR COMMENTS:

In addition to these general comments, the following issues were noted and should be addressed by the authors.

1. There are some inaccuracies/omissions in the Introduction (Page 4) Line 78: ACKRs have not recently emerged – they have been studied for many years since the Duffy antigen was shown to be chemokine binding protein. Lines 80-81: Unlike other ACKRs, ACKR1 does not, to my knowledge, internalise and degrade ligands.

→ **Answer:** *We agree with the reviewer. The text was modified and replaced by “Notably, within this network, a small subfamily of receptors, called atypical chemokine receptors (ACKRs), plays essential regulatory roles. ACKRs bind chemokines without triggering G protein signaling but instead participate in chemotactic events by transporting or capturing the chemokines or internalizing and degrading the ligands in order to resolve inflammatory processes or to shape chemokine gradients”.* (see lines 75 to 79)

MIF is also reported to be an ACKR3 ligand and should be mentioned in the Introduction.

→ **Answer:** *The study by Alampour-Rajabi S et al. (FASEB 2015) reporting MIF as a ligand for ACKR3 was added as suggested (see line 88 and Ref 33).*

2. It would be helpful if there were more complete details included in the Figure Legends. In many cases, it was difficult to find all the information required to understand the Figures (e.g. chemokine concentrations (Fig 1); cell type used (Fig 1); cell ratios (Fig 7E)).

→ **Answer:** *We modified the legends to include more details about the experimental setup (see legends on pages 32 (Fig1), 33 (Fig2), 34 (Fig3), 36-37 (Fig5), 39-40 (Fig6) and 41-42 (Fig7)*

3. Positive controls are required to avoid false negatives in Figure 2 & S2.

→ **Answer:** *In the initial version of the manuscript the following sentence was present in the material and method section to describe the positive control that was used “For library screening experiments, the results are represented as fold over untreated and for each receptor, 100 nM of one known agonist chemokine listed in the IUPHAR repository of chemokine receptor ligands was added as positive control.”*

In order to address the reviewer’s comment, Figures 2 and S2 were modified to show the data of the positive controls. Data are now presented as percentage of positive controls. The value of fold over untreated obtained for each receptor with the corresponding positive control is also presented in the supplementary information (see page 33, supplementary Table 2 and page 3 supplementary information).

4. From Fig S1A and Fig 1A it appears that g-endorphin and big dymorphin might have some effect on ACKR3 and CXCR3, respectively. What statistical tests have been used to exclude these data as significant?

→ **Answer:** *Indeed, we did observe a slight signal increase for big dynorphin on CXCR3 in our initial screen (Figure 1a), as well as later on CX3CR1 and CCR3 (figure 2). To evaluate the relevance of this observation, we performed concentration-response curves of big dynorphin on the three receptors and compared its potency and efficacy with known full agonists of these receptors, CXCL11, CX3CL1*

and CCL13 respectively. These data were present in the initial version of the manuscript in the main text (see lines 152-155 of initial manuscript, now 148-152) and in the supplementary data (Supplementary Figure 2b). For the highest concentration tested (3 μ M), only very low signals were observed, which appeared negligible compared to the positive controls and thus no statistical analysis was performed.

Regarding gamma-endorphin, even though we could detect a weak signal in our initial peptide screening, this hit could not be confirmed in concentration-response curves on ACKR3. Thus, this peptide was not analysed further.

5. Page 7. Lines 132-4. A lot of ligand seems to be required to get full activation, so these data should be discussed in the context of their EC50s and likely physiological relevance.

→ **Answer:** The EC₅₀ values of peptide F ψ G nociceptin 1-13 and the truncated dynorphin variants, dynorphin 2-13 and dynorphin 2-17, listed in lines 132 to 134 (lines 128 to 131 in the revised manuscript) were indeed in low micromolar range (2 to 6 μ M) but all these peptides reached the E_{max} in the concentration range tested (Table 1).

Although it is difficult to judge on the physiological relevance of these values it must be noted that certain opioid peptides such as dynorphin have been shown to reach micromolar to millimolar local concentrations in certain regions of the CNS (Scimemi, A. & Beato, M. *Mol Neurobiol* 40, 289-306(2009), ref 56 in manuscript). Therefore, in this region or particular microenvironment it is conceivable that ACKR3 could be exposed to such high local concentrations of opioid peptides.

A sentence discussing these concerns was added to the discussion (see lines 364 to 367).

6. It is not clear how Figure 3 informs the subsequent parts of the study and the conclusions (lines 178-180) are speculative. It should be included as Supplementary Figure and its description and discussion moved to the Discussion section. Others have done detailed phylogenetic analysis of chemokine receptors (lines 175-178) and this work should be cited (e.g. Nomiyama et al (2011) *Dev Comp Immunol* 35:705).

→ **Answer:** Figure 3 was moved to the supplementary information as suggested (now supplementary Figure 3) and the related text was moved to the discussion and shortened (see lines 404 to 414). The work of Nomiyama et al (2011) *Dev Comp Immunol* was cited (see ref 76)

7. Page 11: Lines 235-6: It would be helpful if suitable publications were cited to highlight those small molecules for ACKR3 modulation that have been developed (e.g. Wang et al (2018). *Front Pharmacol* 9:641).

→ **Answer:** The publication/review of Wang et al (*Front Pharmacol*, 2018) was cited at line 231 together with a recent review of Adlere et al (*Mol Pharmacol*. 2019). See refs 29 and 41

8. The expression of ACKR3 has been examined in the brains of mice and humans in a variety of ways (e.g. Banisadr et al *J Neuroimm Pharmacol* 11:26, Shimizu et al *PLoS One* 6:e20680 and others). The authors should cite this work and should make it more explicit how their data in Fig 7B-C add to current knowledge.

→ **Answer:** *We agree that the expression of ACKR3 in brains of human and mice has been examined by different studies. The goal of the expression analysis in the present study was not to reproduce previous results but to show that ACKR3 is expressed in the same regions of the brain as the classical opioid receptors and to compare their respective expression level in the same analysis. This co-expression/co-localisation and the higher expression of ACKR3 compared to the classical opioid receptors in the different opioid centres of the brain are two important elements supporting our hypothesis that ACKR3 acts as a scavenger receptor regulating opioid peptide availability for the classical opioid receptors.*

We included the suggested references (see refs 52 and 22) and added a sentence (see lines 309) stating that our data are indeed in good agreement with the results from other groups.

9) The authors should use accepted format when referring to genes/transcripts vs proteins i.e. uppercase, lower case, italics etc.. It makes it easier to understand what is being examined.

→ **Answer:** *The formatting of genes/transcripts versus protein was adapted as suggested (see Fig. 7a-c, supplementary Fig. 7) and line 312.*

10) In Figure 7D, instead of using untransfected U87 cells as controls, would it be better to use U87-KOR cells (with no b-arrestin reporter) to allow the authors to see if ACKR3 has more opioid scavenging activity than KOR?

→ **Answer:** *Regarding Figure 7d, now Figure 6c, the main aim of this experiment was to prove that opioid scavenging in U87-ACKR3 cells is exclusively mediated by ACKR3, hence, apparent EC50 values completely shift back to initial levels when ACKR3 is blocked. However, additional uptake experiments/controls with fluorescent ligands were also performed with cells expressing KOR, the classical opioid receptor for dynorphin A (1-13), in comparison to ACKR3-expressing cells to show that although classical opioid receptors internalise opioid peptides, their uptake is far less efficient for the classical opioid receptor than for ACKR3. Indeed, the results show that although KOR has a higher affinity for dynorphin A 1-13 than ACKR3, the uptake of dynorphin-A 1-13 by the cells expressing ACKR3 is significantly higher than that observed with the cells expressing KOR (see lines 271-273 Fig 6b). Similar results were obtained for big dynorphin, the precursor of dynorphin A and dynorphin B, BAM22 and nociceptin. For each of these experiments we compared the uptake/scavenging by ACKR3 and the corresponding classical opioid receptor (KOR, MOR or NOP).*

11) Pages 23-24. The authors point out that datapoints have been removed from the analysis. The full dataset should be presented so that the reader can draw their own conclusions. As I understand it, t-tests are only appropriate test to use when there are two groups in an experiment so an alternative statistical test should be applied to the data in Figure 7D and S6 and the interpretation changed if required.

→ **Answer:** *In the ex vivo rat brain sample analysis, one value, which was > 2SD away from the mean for its experimental group, was excluded. This aberrant value resulted most probably from inadequate storage conditions of dynorphin A used for this particular experiment. The measurement was repeated with a new batch of dynorphin A and the value was consistent with other results*

obtained for the experimental group. Nonetheless, for the sake of transparency, we would like to show to the reviewer the graph with this excluded value:

We also performed a non-parametric Kruskal Wallis test on the entire data set. Despite this obvious outlier, the post hoc Dunn's test still yielded a significant difference between the control and LIH383 3 µM group, with a *p* value of 0.044.

12) Lines 376-377. The work cited does not seem to refer to ACKR3-expressing lymphocytes, monocytes or macrophages, or to the secretion of opioid peptides. Some clarification is required.

→ **Answer:** The text was slightly modified and now reads: "The interplay between ACKR3 and opioids may therefore apply to other physiological systems and in particular the immune system, as subsets of lymphocytes, monocytes or macrophages, proposed to express ACKR3, have also been shown to secrete and respond to opioid peptide". The relevant supporting studies are now correctly referenced (see refs 78-82, lines 416-419).

13) Did the authors use mouse or human chemokines in their study?

→ **Answer:** Unless otherwise stated, for instance in Figure 5e where data were generated with mouse CXCL12 (mCXCL12) and mouse ACKR3 (mACKR3), all the experiments were performed with human receptors, peptides or chemokines. However, most opioid peptides are highly conserved throughout species. For example, human, mouse and rat dynorphin A are identical and thus, the same dynorphin A peptide was used for experiments on human receptors as in rats.

REVIEWERS' COMMENTS:

Reviewer #1 (Remarks to the Author):

My concerns have been fully addressed. I look forward to seeing the paper on line.

Reviewer #2 (Remarks to the Author):

The authors have produced a revised version of their manuscript that includes a lot of new data and which has comprehensively and successfully addressed all the issues raised in the first review. The authors are to be commended for their thoroughness and attention to detail, and for producing what is now an excellent piece of work that will be of broad interest to readership of Nat Comms and likely to be widely cited in the future.